# Pathogenic *UNC13A* variants cause a neurodevelopmental syndrome by impairing synaptic function

The *UNC13A* gene encodes a presynaptic protein that is crucial for setting the strength and dynamics of information transfer between neurons. Here we describe a neurodevelopmental syndrome caused by germline coding or splice-site variants in *UNC13A*. The syndrome presents with variable degrees of developmental delay and intellectual disability, seizures of different types, tremor and dyskinetic movements and, in some cases, death in early childhood. Using assays with expression of *UNC13A* variants in mouse hippocampal neurons and in *Caenorhabditis elegans*, we identify three mechanisms of pathogenicity, including reduction in synaptic strength caused by reduced UNC13A protein expression, increased neurotransmission caused by UNC13A gain-of-function and impaired regulation of neurotransmission by second messenger signalling. Based on a strong genotype–phenotype-functional correlation, we classify three UNC13A syndrome subtypes (types A–C). We conclude that the precise regulation of neurotransmitter release by UNC13A is critical for human nervous system function.

UNC13A, the major UNC13 paralog in mammals, is a highly conserved presynaptic protein that is essential for chemical synaptic transmission. It is required for assembly of the SNARE complex that bridges between neurotransmitter-filled synaptic vesicles (SVs) and the plasma membrane, placing SVs in a fusion-competent, 'primed' state[1–5]. UNC13A expression levels are positively correlated with the size of the readily releasable SV pool (the RRP[6]) and with the signaling strength of synapses[7–9]. During periods of high-frequency synaptic activity, UNC13A supports the resupply of SVs to counterbalance SV consumption[10–14], sustaining neurotransmission. UNC13A activity is enhanced through direct binding of $Ca^{2+}$, $Ca^{2+}$-calmodulin, $Ca^{2+}$-phospholipids and diacylglycerol (DAG) to specific regulatory domains. The acceleration of UNC13A activity results in dynamic changes in neurotransmission strength, a phenomenon known as synaptic plasticity[10–13,15,16].

In rodents, Unc13a is expressed in the vast majority of neuronal subtypes and in some cells of neural lineage[17,18]. Most neuronal subtypes additionally express the paralogs Unc13b and/or Unc13c[17], which have similar functions. Mice that constitutively lack Unc13a die shortly after birth[19,20] owing to a 90% loss of the RRP at excitatory synapses[19]. The combined elimination of Unc13a/Unc13b results in a complete block of neurotransmission at both excitatory and inhibitory synapses owing to complete RRP loss[20]. Consequently, newborn Unc13a/b double knockout (DKO) mice are paralyzed but have intact nervous system development[20,21].

A pivotal role for UNC13A in human neurological disorders is emerging. Two patients harboring homozygous truncating variants in *UNC13A* showed profound developmental delay and died during early childhood[22,23]. Another patient harboring a de novo missense variant (P814L) in *UNC13A* exhibited mild intellectual disability, dyskinesia and intention tremor[24]. Certain deep intronic single-nucleotide polymorphisms contribute to genetic risk for the neurodegenerative conditions amyotrophic lateral sclerosis (ALS) and frontotemporal dementia (FTD)[25–27], and promote ALS/FTD disease progression[26–33]. Recent studies identified reduced UNC13A expression in neurons with ALS/FTD pathology owing to TDP-43-mediated alterations in splicing[34,35], but the mechanisms linking reduced UNC13A expression to disease progression remain unknown[33,36].

---

✉e-mail: R.Asadollahi@greenwich.ac.uk; Lipstein@FMP-berlin.de

We identified a neurodevelopmental syndrome caused by variants in the *UNC13A* gene. Systematic patient and variant characterization enabled classification of three disease subtypes. Patients with a type A condition present with profound global developmental delay (GDD) and early-onset seizures. These patients harbor biallelic loss-of-function missense, truncating or splice-site variants that lead to a >50% reduction of UNC13A expression and to a severe reduction in neurotransmission strength in experimental models. Patients with a type B condition exhibit developmental delay, particularly speech delay, and ataxia, tremor or dyskinetic movements as hallmarks of the condition. These patients harbor de novo missense variants that result in a gain of UNC13A function, leading to enhanced neurotransmission. The type C condition is caused by a familial heterozygous missense variant that results in altered regulation of UNC13A function. The patients are mildly affected, exhibiting learning difficulties to mild–moderate intellectual disability and seizures.

Overall, we demonstrate that *UNC13A* disease-causing variants operate in a cell-autonomous fashion to change synaptic strength and plasticity. Importantly, we present evidence that reduction of UNC13A protein expression to 20–30% of wild-type (WT) levels strongly impairs synaptic transmission and plasticity and has severe consequences in humans, a finding that may be important to understand the role of UNC13A in ALS and FTD. The mechanisms identified here may inform the development of therapeutic approaches.

## Results

### A neurodevelopmental syndrome caused by pathogenic *UNC13A* variants

We identified diverse *UNC13A* variants in 48 index patients presenting with neurodevelopmental deficiencies (Supplementary Data 1). Based on genotype–phenotype-functional assessment, we classified variants in 20 patients as pathogenic (Fig. 1a) and in eight patients as (likely) benign. Variants in 20 patients remained of uncertain significance (Supplementary Data 1 and Extended Data Fig. 1).

A first group of six patients (18 months to -15 years old; Fig. 1a,b (magenta), Supplementary Data 1 and Extended Data Figs. 2 and 3) presented with severe-to-profound GDD or intellectual disability, hypotonia and seizures of different types (largely controllable with medication) or death in early childhood caused by respiratory failure after pneumonia in one case. *UNC13A* variants in these patients were homozygous or compound heterozygous missense, insertion–deletion or splice-site variants with gene-disrupting splicing effects proven by minigene assays (Supplementary Data 1 and Extended Data Figs. 2 and 3). Notably, parents of patients with biallelic variants in this study who were heterozygous carriers of loss-of-function missense or gene-disrupting splice-site variants, and parents of an individual in a previously published case[22] with a homozygous gene-disrupting

variant were all reportedly healthy, demonstrating possible tolerance to heterozygous loss-of-function *UNC13A* variants, as has also been shown in mice[19].

A second group of 13 patients (21 months to 32 years old; Fig. 1a,b (black) and Supplementary Data 1) harbored pathogenic, heterozygous de novo missense variants with multiple substitutions at amino acids 808, 811 and 814. They presented with variable degrees of GDD, hypotonia, seizures of different types (mainly refractory to treatment) and typically exhibited ataxia, tremor or dyskinetic movements rarely observed in other patients (Supplementary Data 1 and Supplementary Videos 1 and 2). The protein region spanning the seven highly conserved amino acids 808–814 forms a hotspot for an autosomal dominant pattern of inheritance and is predicted to be highly intolerant to variation based on the MetaDome database[37] and AlphaMissense scores[38] (Fig. 1c,d). We term this region the 'UNC13 hinge' because it links a regulatory domain cluster with the MUN domain, which mediates SNARE complex assembly[39,40] (Fig. 1b).

The third group of patients consisted of a family with at least four affected members across two generations (4 years to 35 years old; Fig. 1a,b (brown) and Supplementary Data 1) harboring a pathogenic heterozygous missense variant (C587F) that caused learning difficulties to mild–moderate intellectual disability as well as controlled seizures. The relatively mild presentation of this variant led to autosomal dominant familial heritability.

Altogether, pathogenic variants in *UNC13A* cause a spectrum of neurodevelopmental deficiencies (Fig. 1e; mainly GDD or intellectual disability, hypotonia, seizures and abnormal movement features) with both autosomal dominant (de novo or inherited) and recessive inheritance, depending on the type and location of the variants. Additionally, we identified de novo heterozygous or biallelic variants of uncertain significance scattered throughout the UNC13A protein sequence (Extended Data Fig. 1 and Supplementary Data 1). Although many of these variants are high-level (hot) candidates for impacting protein function based on in silico predictions (Fig. 1d and Supplementary Data 1), reliable readouts proving their pathogenicity are currently unavailable.

### Characterization of UNC13A variant pathogenicity

UNC13A is a multidomain protein (Fig. 1b) impacting diverse synaptic transmission properties[7,10–15,19,24]. We selected a subset of variants with substitutions at key UNC13A domains (Fig. 2a) and comprehensively characterized their impact on synapse function. We show that variants E52K, R202H, C587F, G808D/C and P814L are pathogenic, while our assays did not detect phenotypic changes for R799Q and N1013S, which we therefore classify as (likely) benign (Supplementary Data 1–3 and Extended Data Fig. 4). A comparative summary of all genetic, clinical and experimental data is presented in Extended Data Fig. 5.

**Fig. 1 | Landscape of *UNC13A* patient variants identified in this study.**
**a**, Location of validated pathogenic variants identified in 20 patients (Supplementary Data 1) mapped on a schematic representation of the *UNC13A* gene (NM_001080421.2). Variants with biallelic inheritance are in magenta (p.T117Rfs*18 and p.R202H, and c.767+1 G > T and c.4073+1 G > A are in a compound heterozygous state; others are in a homozygous state), heterozygous de novo variants are in black and a heterozygous, inherited variant is in brown. Variants p.G808D and p.P814L were detected in four and six patients, respectively. Exon colors correspond to the color of protein domains in **b**.
**b**, Validated pathogenic missense variants overlaid on a schematic representation of the UNC13A protein. The sequence of the hotspot region, termed here as the 'UNC13 hinge', is magnified. **c**, *UNC13A* tolerance landscape generated based on the observed missense and synonymous variants in gnomAD and corrected for the sequence composition (MetaDome database[37]). The degree of tolerance for missense variants is color-coded (lowest scores, red, highly intolerant; highest scores, blue, highly tolerant). **d**, AlphaMissense scores[38] plotted for all missense variants (pathogenic, of uncertain significance (VUS) and (likely)

benign) reported in this study (Supplementary Data 1; heterozygous de novo missense variants, bold typeface; heterozygous inherited or biallelic variants, non-bold typeface; validated pathogenic variants, red; VUS, gray; validated (likely) benign, blue). Notably, all validated pathogenic variants reported in this study are within the pathogenic AlphaMissense score range, and three out of four validated benign variants fall within the benign score range. We consider VUS variants in the pathogenic AlphaMissense score range as hot VUS (14 variants) and those within the benign/ambiguous score range as cold VUS (12 variants). **e**, Left: age of patients (with median and interquartile range; error bars, s.d.) at the latest physician visit, stratified by inheritance pattern. Middle: major clinical features of 20 patients with pathogenic variants (Supplementary Data 1), stratified by inheritance pattern. The presence or absence of features is shown by the indicated colors together with the respective percentage for different inheritance patterns. Right: standard deviation scores for growth parameters of patients with pathogenic variants at birth and at the latest visit are illustrated by inheritance pattern (HC, head circumference).

## UNC13A amino-terminal disease variants reduce UNC13A protein levels

Neurodevelopmental disorders have been linked to deficiencies in synapse development[41]. To examine whether *UNC13A* variants affect synaptogenesis or presynaptic UNC13A abundance, we cultured hippocampal neurons from Unc13a/b DKO mice in a microisland culture system[42,43] (Fig. 2b), where single neurons grow on astrocytic islands, making synapses onto themselves ('autapses'). We re-introduced WT rat *Unc13a* cDNA (UNC13A[WT]; ~94% identity with human UNC13A) or *UNC13A* cDNA encoding disease-related variants using lentiviral transduction[44] at day

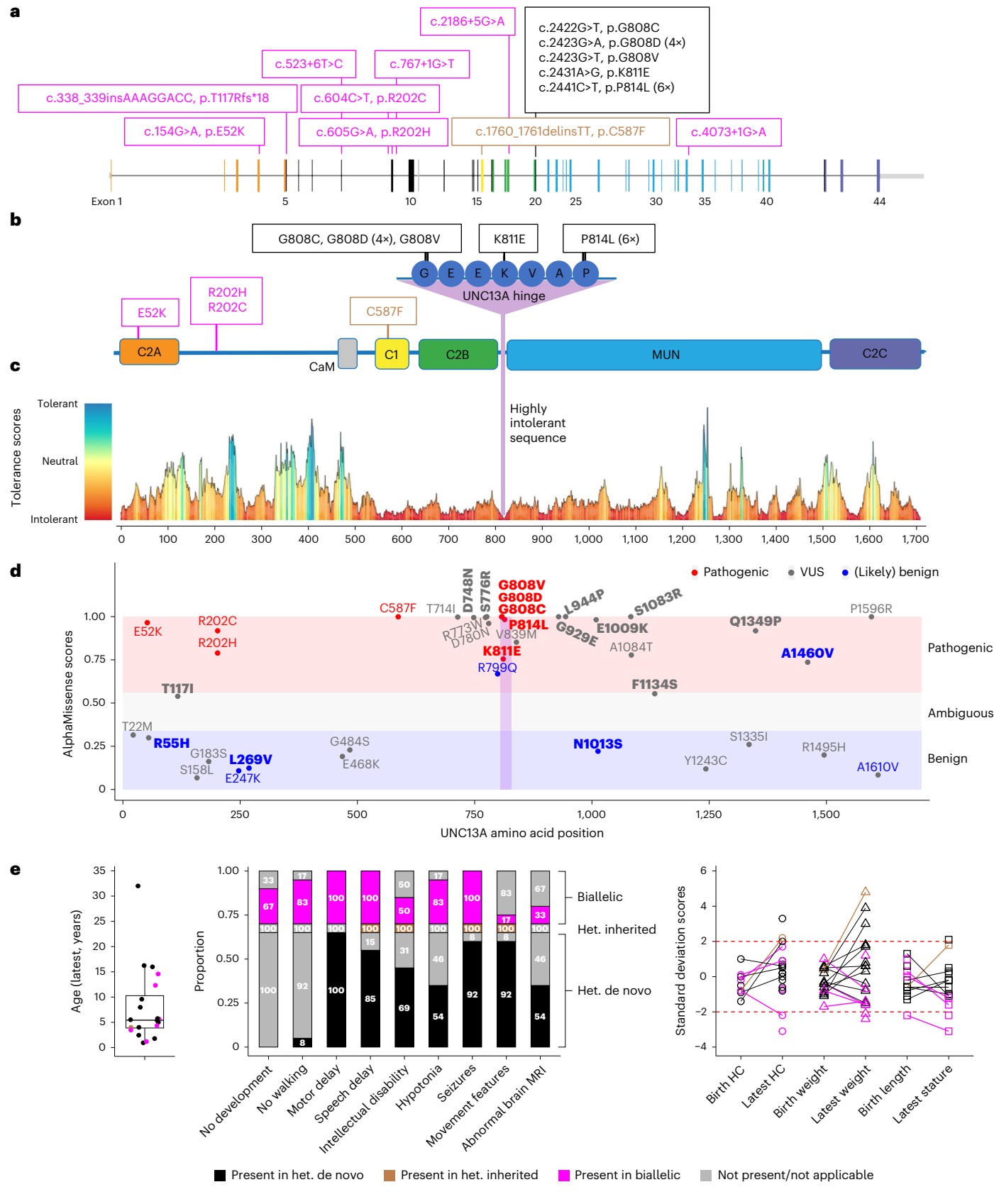

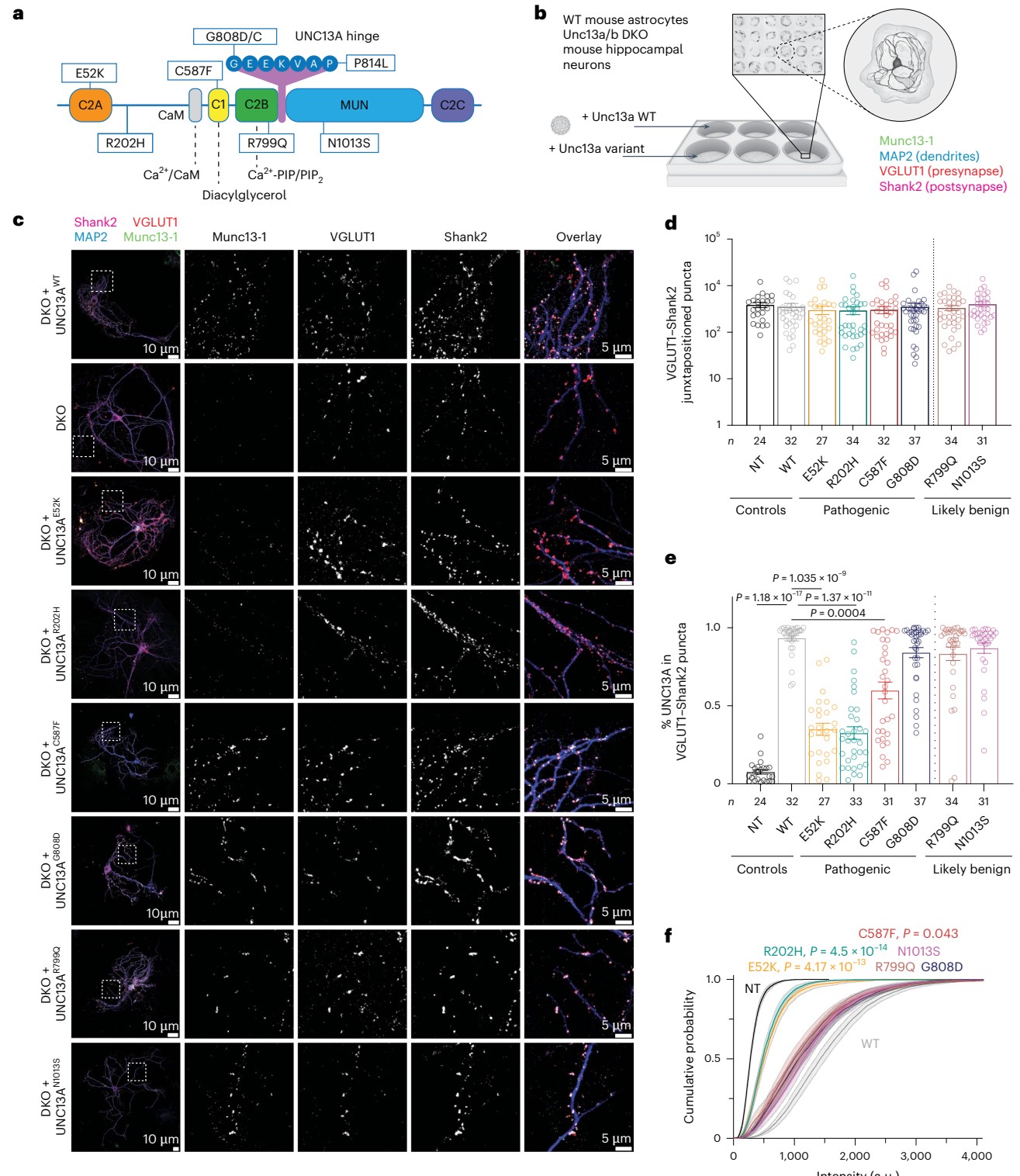

**Fig. 2 | UNC13A disease-causing variants change UNC13A expression levels at synapses. a**, Location of studied variants within the UNC13A protein domains (not drawn to scale). **b**, Schematic representation of the autaptic cell culture system used in this study. **c**, Left: example images of cultured autaptic hippocampal neurons from Unc13a/b DKO mice that were transfected on DIV 2–3 with UNC13A–GFP cDNA encoding WT or disease-related variants and stained with antibodies against the presynaptic marker VGLUT1, the postsynaptic marker Shank2, the dendritic marker MAP2 and with an antibody against GFP (*n* = 24–37). Right: magnification of the regions indicated by the white boxes in the merged image. **d**, Quantification of VGLUT1–Shank2 juxtapositioned puncta per neuron. NT, not transfected. **e**, Fraction of VGLUT1–Shank2 juxtapositioned puncta where UNC13A–GFP co-staining was detected. Bars and circles in **d** and **e** represent means and values for individual neurons, respectively; error bars, s.e.m. The *P* values in **e** refer to pairwise comparisons of mean fractions with WT using Dunn's multiple comparison test. **f**, Cumulative probability distributions of UNC13A–GFP intensities at VGLUT1–Shank2 juxtapositioned puncta, reporting relative UNC13A expression levels at individual synapses. For each cell analyzed, the mean Munc13 intensity over all synapses was obtained and, subsequently, the pooled mean was calculated for each genotype (Supplementary Data 2). a.u., arbitrary units. The *P* values in **f** refer to pairwise comparisons of mean intensities with WT using Tukey's multiple comparisons test. See Supplementary Data 2 for further details.

in vitro (DIV) 2–3. Following immunocytochemical labeling, synapses were defined as VGLUT1-positive puncta (a presynaptic marker) colocalizing with Shank2-positive puncta (a postsynaptic marker). We found no significant change in the number of formed synapses between the different conditions (Fig. 2c,d and Supplementary Data 2). We then quantified the fraction of VGLUT1–Shank2 puncta exhibiting UNC13A immunoreactivity (Fig. 2c,e and Supplementary Data 2). UNC13A[WT] was detected in 93.4 ± 1.75% of the VGLUT1–Shank2-positive puncta, and similar colocalization levels were detected for UNC13A with variants in the Ca$^{2+}$-phospholipid binding C2B domain (UNC13A[R799Q]), UNC13A hinge (UNC13A[G808D]) and MUN domain (UNC13A[N1013S]). Neurons expressing a UNC13A variant in the DAG-binding C1 domain (UNC13A[C587F]) exhibited a mild but statistically significant reduction in the degree of colocalization. Remarkably, neurons expressing either of the two amino-terminal variants, UNC13A[E52K] and UNC13A[R202H], displayed a strong reduction in the fraction of synapses co-labeled for UNC13A (35 ± 3.9% and 32.5 ± 4% of WT levels, respectively). Moreover, substantially weaker UNC13A labeling intensities were evident (UNC13A[E52K], 23 ± 11% of WT levels; UNC13A[R202H], 20 ± 13% of WT levels; Fig. 2f and Supplementary Data 2). Taken together, UNC13A disease variants do not affect synapse number in vitro, but the E52K and R202H substitutions cause a strong reduction of synaptic UNC13A levels.

### UNC13A amino-terminal disease variants cause reduced neurotransmission

E52 is located at the interaction interface between the UNC13A C2A domain and RIMS1/RIMS2 (refs. 45–48) (Fig. 3a,b), while R202 is part of an intrinsically disordered region (IDR; amino acids 151 to ~450; Fig. 1a). The E52K and R202H disease variants are inherited in an autosomal recessive manner and cause a severe disorder (Fig. 3c,d). Expression of UNC13A[E52K] or UNC13A[R202H] in HEK293 cells, which lack UNC13A but express the misfolded protein detection machinery, showed no evidence that these variants cause inherent protein instability (Extended Data Fig. 6).

To assess how variants affect synaptic transmission, we conducted whole-cell voltage-clamp recordings in autaptic Unc13a/b DKO glutamatergic hippocampal neurons (Fig. 3e)[20,42] transfected with UNC13A[WT] or UNC13A carrying disease variants. In UNC13A[E52K]-expressing neurons (Fig. 3f), we could not detect spontaneous miniature excitatory postsynaptic currents (mEPSCs) or action potential (AP)-evoked excitatory postsynaptic currents (eEPSCs; Fig. 3g–j and Supplementary Data 3). Application of hypertonic sucrose solution to assess the size of the RRP[6] revealed the absence of a measurable RRP in UNC13A[E52K] neurons, indicative of a severe SV priming deficit (Fig. 3k,l). Given that high-frequency AP firing can enhance UNC13A activity, we delivered trains of 40 APs at 40 Hz frequency but did not detect synaptic currents throughout the trains (Fig. 3m). Together, we conclude that UNC13A[E52K] does not rescue synaptic transmission to measurable levels.

By contrast, UNC13A[R202H] expression (Fig. 3n) rescued synaptic transmission, albeit with lower efficacy than UNC13A[WT]. Mean frequency and amplitude of mEPSCs were similar to those measured in neurons expressing UNC13A[WT] (Fig. 3o–q), but the average eEPSC size (Fig. 3r,s) and RRP charge (Fig. 3t,u) were both reduced by ~40%. The mean vesicular release probability ($\bar{p}_{vr}$), calculated as the charge transfer during a single AP-induced eEPSC divided by the charge transfer in response to sucrose-evoked RRP release, was unaltered (Fig. 3v).

Taken together, these results demonstrate that two variants cause lower synaptic UNC13A abundance and reduced synaptic strength in mouse neurons. Affected patients presented with profound GDD and no motor development (Figs. 1e and 3c), consistent with decreased synaptic efficacy observed in vitro. Interestingly, although both substitutions reduce UNC13A levels to similar extents (Fig. 2e,f), UNC13[R202H] expression partially rescues synaptic transmission, while UNC13[E52K] expression does not. This probably reflects the combined effect of lower UNC13A protein levels and functional consequences of the E52K

exchange on the C2A domain function[47]. The substantial neurotransmission in UNC13A[R202H]-expressing neurons indicates that very low UNC13A levels are sufficient for synapse function.

### UNC13A hinge variants cause a gain-of-function in neurotransmission

We identified 13 patients with de novo variants in the UNC13A hinge (Fig. 4a–d). We previously reported one patient with an UNC13A P814L variant[24], causing a gain-of-function in synaptic transmission. To consolidate variant pathogenicity and establish functional causality between the expression of hinge variants and synaptic transmission properties, we studied the frequently affected residue G808 (Fig. 4a,d).

In autaptic hippocampal Unc13a/b DKO neurons expressing UNC13A[G808D]–GFP, we observed a greater than twofold increase in the frequency of mEPSCs compared to neurons expressing the WT variant, whereas mEPSC amplitudes were unaltered (Fig. 4e–g). Given that larger RRP size could underlie an increased mEPSC frequency, we estimated the RRP sizes and found similar values in UNC13A[G808D]-expressing and UNC13A[WT]-expressing neurons (Fig. 4h,i). Together, these findings are consistent with the notion that RRP SVs in resting UNC13A[G808D]-expressing neurons are more likely to fuse spontaneously, possibly indicating higher SV fusogenicity[49] or a larger fraction of 'tightly-docked' SVs[50]. UNC13A[G808D]-expressing neurons also showed significantly larger eEPSC charges (Fig. 4j,k). Given that the average mEPSC charge and RRP size were unaltered (Fig. 4f,i), the increased eEPSC charge is indicative of a larger fraction of the RRP released by single APs (higher $\bar{p}_{vr}$; Fig. 4l). We conclude that UNC13A[G808D] leads to a gain-of-function in spontaneous and evoked neurotransmission.

Given that all UNC13A hinge variants were heterozygous, we tested whether a gain-of-function phenotype prevails in the presence of a WT allele. We generated homozygous and heterozygous *C. elegans* strains harboring the G808C and P814L substitutions using CRISPR–Cas9 targeting of the *unc-13* gene. The *C. elegans* and human UNC13A protein products are highly homologous (~50% identity, ~70% sequence similarity) and serve similar functions[51–54]. As a behavioral readout of synaptic function, we assayed loss of movement in response to exposure to the acetylcholinesterase inhibitor aldicarb (Fig. 4m), with faster paralysis indicating acetylcholine hypersecretion. Compared with WT worms, paralysis occurred significantly faster and to a similar extent in homozygous and heterozygous G808C and P814L mutant worms (Fig. 4n,o (green and blue)). These data are consistent with a gain-of-function in acetylcholine secretion, also in the presence of a WT allele.

Based on these results, we conclude that UNC13A hinge variations increase the degree of spontaneous and evoked neurotransmission. Given the substantial number of patients, and because the functional phenotypes of neurons expressing UNC13A[P814L] and UNC13A[G808D] in mouse neurons and G808C and P814L in *C. elegans* are nearly identical (Extended Data Fig. 5), we conclude that UNC13A hinge variations are pathogenic and act through a similar mechanism.

### A familial variant in the UNC13A C1 domain

We identified four patients harboring a heterozygous missense variant (c.1760_1761delinsTT; C587F) in the UNC13A DAG-binding C1 domain (Fig. 5a–d)[15,55]. In cultured WT synapses, the membrane-permeable DAG analog Phorbol 12,13-dibutyrate (PDBu) causes a greater than twofold increase in synaptic strength within seconds after application[15,56]. To test whether C587F hampers DAG sensing, we used a protein-translocation assay. In HEK293 cells, overexpressed UNC13A[WT]–GFP undergoes translocation to the plasma membrane upon PDBu application (Fig. 5e,f). A C1 domain point mutation, H554K, blocks this translocation[14] (Fig. 5e,f). We found that the disease variant C587F blocks UNC13A translocation, while other variants in this study exhibit WT-like behavior (Fig. 5e). We then characterized synaptic transmission and DAG responsiveness in autaptic neurons expressing UNC13A[C587F] (Fig. 5g–n and

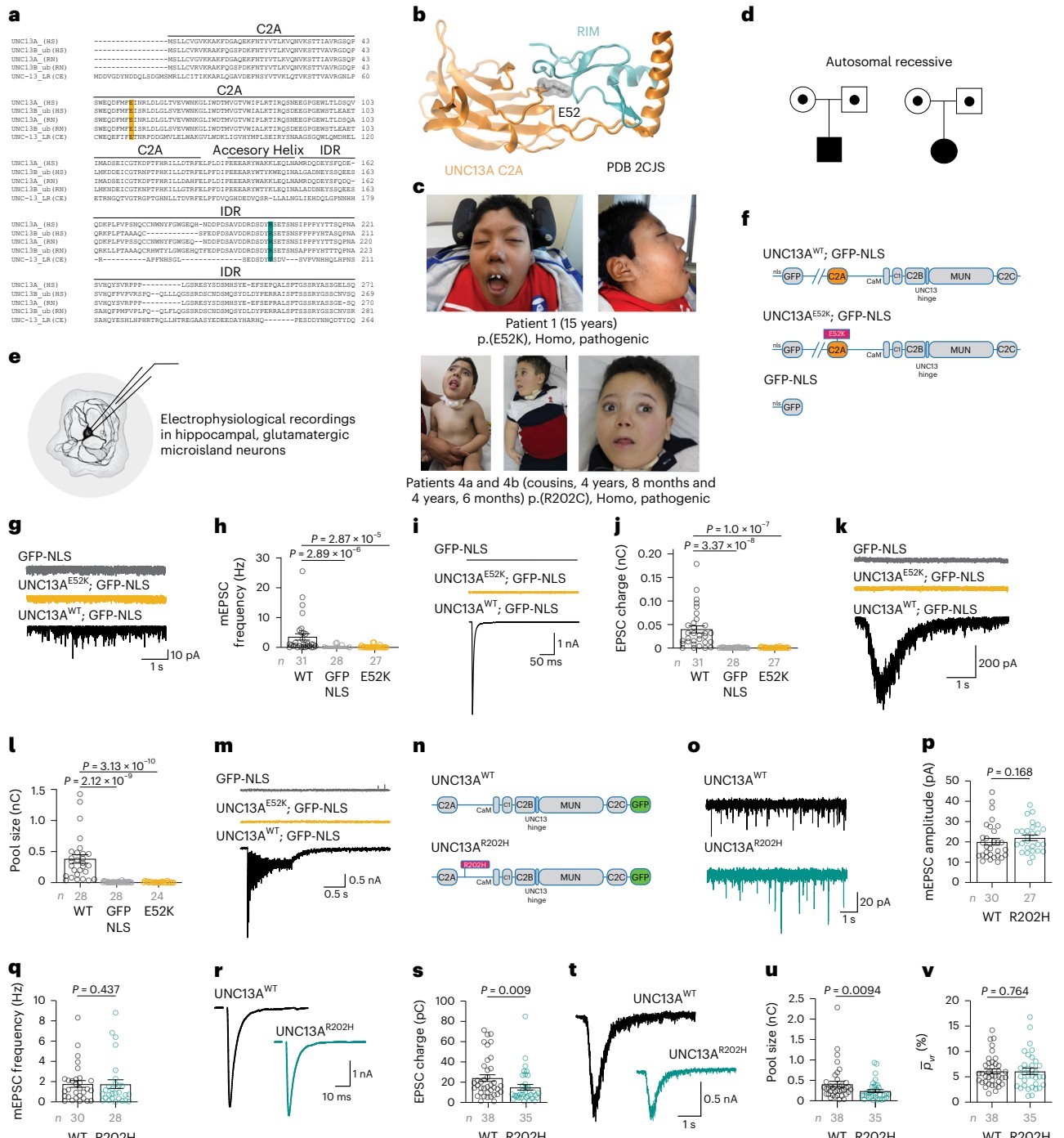

**Fig. 3 | Amino-terminal disease variants abate synaptic strength. a**, Amino acid sequence alignment of the C2A domain and adjacent region for UNC13A homologs in different species (disease variants are highlighted by a bar; E52K, orange; R202H, green; HS, *Homo sapiens*; RN, *Rattus norvegicus*; CE, *C. elegans*). **b**, Structure of the UNC13A C2A domain (orange) in complex with the RIM zinc-finger domain (blue) (PDB 2CJS)[72]. E52 is shown in a space-filled presentation. **c**, Pictures of three patients with pathogenic homozygous amino-terminal variants, showing narrow forehead, highly arched eyebrows, impression of hypertelorism, depressed nasal bridge, small nasal tip and underdeveloped, anteverted ala nasi, long philtrum, impression of retrognathia and large, mildly forward-facing earlobes. **d**, Representative pedigrees of patients with pathogenic biallelic amino-terminal variants, illustrating an autosomal recessive pattern of inheritance with apparently healthy carrier parents. **e**, Schematic illustration of electrophysiological recordings in autaptic neurons. **f**, Constructs used in electrophysiological experiments in **g**–**m** (NLS; nuclear localization signal). **g**–**m**, Characterization of synaptic transmission in autaptic hippocampal

neurons expressing UNC13A^WT (black), UNC13A^E52K (orange) or GFP-NLS (gray). Example traces (**g**) and summary data showing mean frequencies (**h**) of mEPSCs. Example traces (**i**) and summary data showing mean charges (**j**) of eEPSCs. Example traces (**k**) and summary data showing mean charges (**l**) of sucrose-evoked EPSCs as a measure of RRP size. No eEPSCs were elicited in UNC13A^E52K-expressing neurons during high-frequency AP trains (40 Hz) (**m**). **n**, The constructs used in electrophysiological experiments in **o**–**v**. Example traces (**o**), mean amplitudes (**p**) and mean frequencies (**q**) of spontaneously occurring mEPSCs in neurons expressing UNC13A^WT (black) and UNC13A^R202H (green). **r**,**s**, Example eEPSC traces (**r**) and mean eEPSC charge (**s**). Smaller sucrose-evoked EPSCs (**t**) indicate a reduced RRP size in UNC13A^E52K-expressing neurons (**u**), while the mean vesicular release probability was unchanged (**v**). Data were obtained from at least three independent cultures per condition. Statistical analysis: Kruskal–Wallis test (**g**–**m**) and two-tailed Mann–Whitney test (**o**–**v**). In **h**,**j**,**l**,**p**, **q**,**s**,**u** and **v**, bars and circles represent means and values for individual neurons, respectively; error bars, s.e.m. See Supplementary Data 3 for further details.

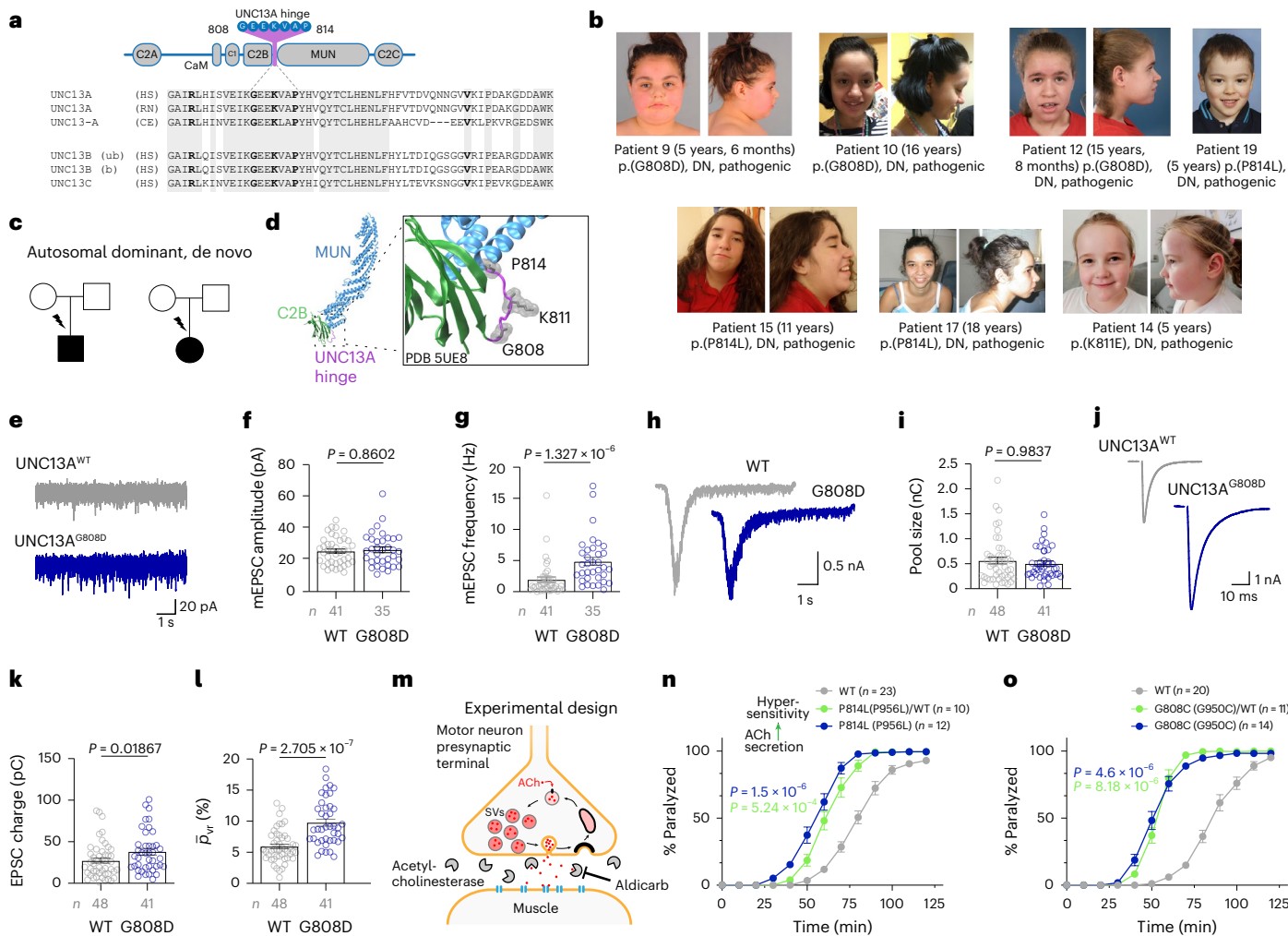

**Fig. 4 | The UNC13A hinge region is a disease hotspot. a**, A sequence alignment of UNC13A homologs illustrating that the UNC13 hinge domain is conserved (residues found mutated in patients in bold). **b**, Facial portraits of patients with pathogenic de novo hinge variants, displaying high anterior hairline with an appearance of tall forehead, impression of hypertelorism, deeply set eyes, attached earlobes, smooth, rather short philtrum, thin upper lip vermilion, everted lower lip vermilion, bulbous nasal tip and retrognathia. **c**, Representative pedigrees of patients with pathogenic de novo variants, illustrating an autosomal dominant pattern of inheritance. **d**, Crystal structure of the UNC13A (PDB 5UE8) identifies the UNC13A hinge as non-structured (purple), connecting two structured domains. **e–l**, Characterization of synaptic transmission in autaptic hippocampal neurons expressing UNC13A^WT (gray) or UNC13A^G808D (blue). Example traces (**e**) and summary data showing mean amplitudes (**f**) and mean frequencies (**g**) of mEPSCs. Example traces (**h**) of sucrose-evoked EPSCs and summary plots of the measured RRP size (**i**). Example traces (**j**) of eEPSCs and summary plots of eEPSC charge (**k**) and vesicular release probability (**l**). **m**, Schematic illustration of the aldicarb assay in *C. elegans*. Aldicarb-induced accumulation of ACh in the extrasynaptic space leads to receptor desensitization and to worm paralysis. Synaptic phenotypes associated with ACh hypersecretion lead to faster ACh accumulation and to faster paralysis. Reduced secretion results in slower paralysis. **n**, CRISPR-generated heterozygous (green) and homozygous (blue) knock-in UNC-13^P814L (UNC-13^P956L) worms exhibit faster aldicarb-induced paralysis than WT worms. **o**, Similar results obtained in heterozygous (green) or homozygous (blue) UNC-13^G808C (UNC-13^G950C) knock-in worms. Statistical analysis was performed by using a two-tailed Mann–Whitney test (**f**, **g**, **i**, **k**, **l**) or Kruskal–Wallis test followed by Dunn's test (**n**, **o**). Bars and circles in **f**, **g**, **i**, **k** and **l** represent means and values for individual neurons, respectively; error bars, s.e.m. Data were obtained from at least three independent cultures per condition. Circles in **n** and **o** represent averaged data from >10 worms. See Supplementary Data 3 for further details.

Supplementary Data 3). Basal spontaneous and AP-evoked synaptic transmission were largely similar in Unc13a/b DKO neurons expressing UNC13A^C587F or UNC13A^WT. This is consistent with a similar time course of aldicarb-induced paralysis in homozygous C587F worms and WT worms (Fig. 5o). However, PDBu failed to potentiate eEPSC amplitudes in autaptic neurons (Fig. 5p–r), in agreement with the absence of PDBu-induced translocation (Fig. 5f). We conclude that the UNC13A^C587F variation acutely changes the responsiveness of UNC13A to DAG signaling, while not strongly interfering with basal synaptic transmission (see Extended Data Fig. 7 for molecular dynamics simulation data).

Finally, we tested whether other disease variants impair the PDBu-induced synaptic potentiation. Expression of UNC13A^G808D, with a presumably intact C1 domain (Fig. 5e), resulted in little PDBu-induced

potentiation (~20% of WT; Fig. 5s). Conversely, neurons expressing R202H exhibited a tendency towards stronger PDBu-induced potentiation than UNC13A^WT-expressing neurons (Fig. 5t). This is probably a consequence of already altered basal synaptic transmission properties: synaptic strength is enhanced in neurons expressing UNC13A^G808D, which may partially occlude PDBu potentiation. By contrast, neurons expressing UNC13A^R202H exhibit lower basal synaptic strength, which may allow stronger potentiation. These data suggest an unbalanced DAG signaling in UNC13A^G808D-expressing and UNC13A^R202H-expressing neurons.

### UNC13A variants alter short-term synaptic plasticity
During repetitive activation, synapses exhibit short-term plasticity (STP)[57], a use-dependent alteration of synaptic strength on a time scale

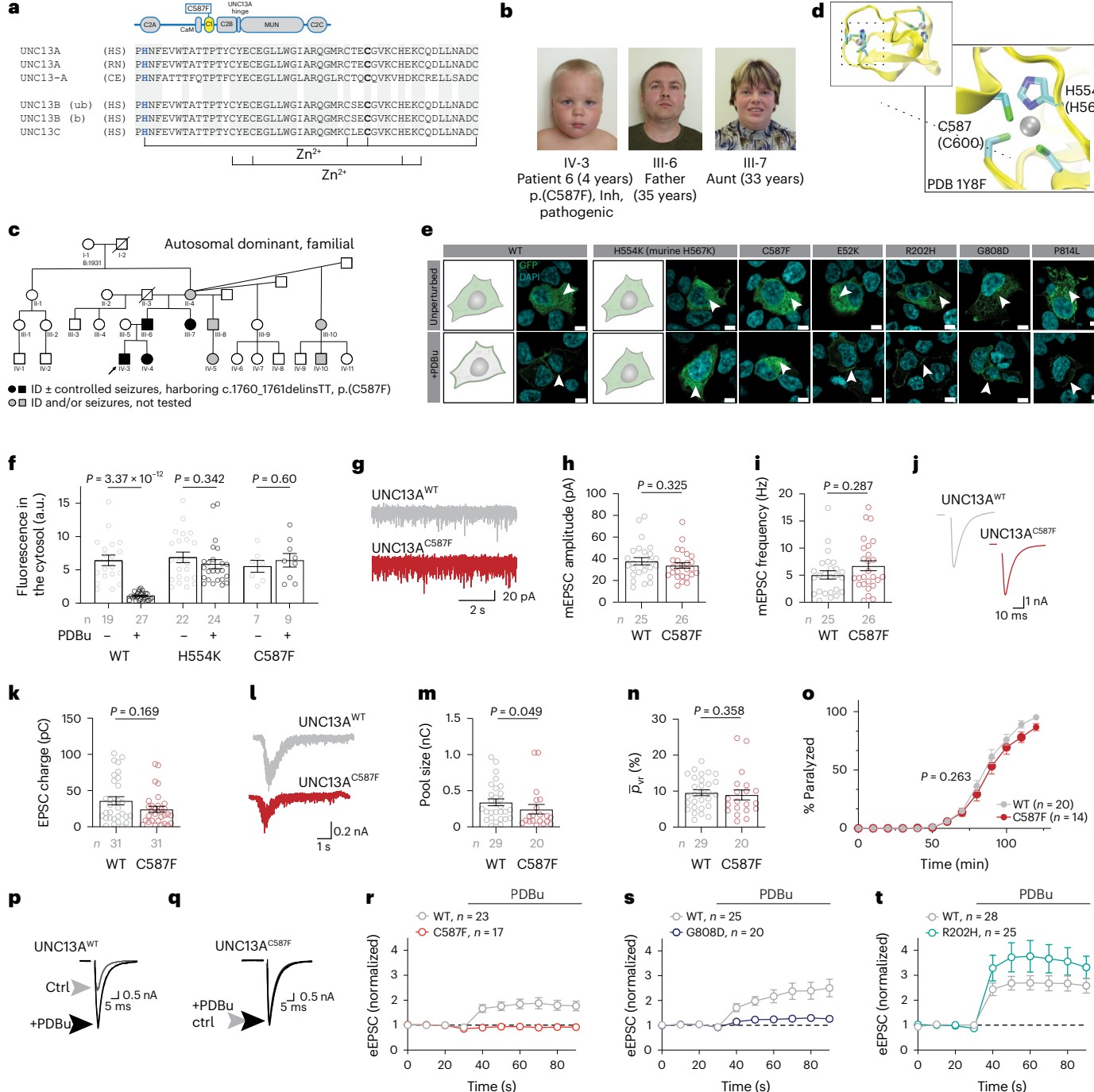

**Fig. 5 | UNC13 disease variations change the sensitivity of synaptic transmission to second messenger regulation by the DAG pathway.**
**a**, Sequence alignment for UNC13 paralogs highlighting H554 (blue) and C587 (bold). Lines below indicate $Zn^{2+}$-coordinating residues. **b**, Facial portraits of index patient 6, his father and aunt, carrying the pathogenic variant, displaying a high, broad forehead, impression of hypertelorism, deeply set eyes, fullness of the upper lateral eyelid, depressed nasal bridge, underdeveloped ala nasi and small nasal tip, smooth, broad philtrum and everted vermilion of the upper lip. **c**, A four-generation pedigree, with four individuals confirmed as harboring c.1760 G > T, p.(C587F). **d**, The C1 domain structure (top) and a zoomed-in view of one $Zn^{2+}$-binding pocket. The corresponding amino acid numbers in the rat UNC13A are given in brackets. **e**, HEK293FT cells transiently transfected with UNC13A–GFP constructs encoding WT or patient variants, treated with 1 μM PDBu for 1 h (scale bar, 5 μm; n = 10–15 per condition; white arrowheads highlight UNC13A location (green)). **f**, Quantification of the cytosolic GFP signal in the presence or absence of PDBu. **g**–**n**, Characterization of synaptic transmission in autaptic hippocampal neurons expressing UNC13A^WT (gray) or UNC13A^C587F (red). Example traces (**g**) and summary plots showing mean amplitude (**h**) and mean mEPSC frequency (**i**). Example traces (**j**) and summary plots showing mean eEPSC charge (**k**). Example traces (**l**) and summary plots showing mean sucrose-evoked EPSC charge (**m**). **n**, Quantification of vesicular release probability. **o**, Time to paralysis in CRISPR-targeted *C. elegans* worms carrying a homozygous C587F (C729F) variation and exposed to aldicarb. **p,q**, Example eEPSCs before and after application of PDBu in neurons expressing UNC13A^WT (**p**) or UNC13A^C587F (**q**). **r**, The average time courses of eEPSC amplitude change. **s,t**, Similar data for UNC13A^G808D (**s**, blue) or UNC13A^R202H (**t**, green). In **o** and **r**–**t**, circles represent an average from >10 worms or an average of the indicated number of neurons. Bars and circles in **f**, **h**, **i**, **k**, **m** and **n** represent means and values for individual neurons, respectively. Data were obtained from at least three independent cultures per condition; error bars, s.e.m. Statistical analysis was performed by using a two-tailed Mann–Whitney test (**f**, **h**, **i**, **k**, **m**, **n**) or Wilcoxon test (**o**). See Supplementary Data 3 for further details.

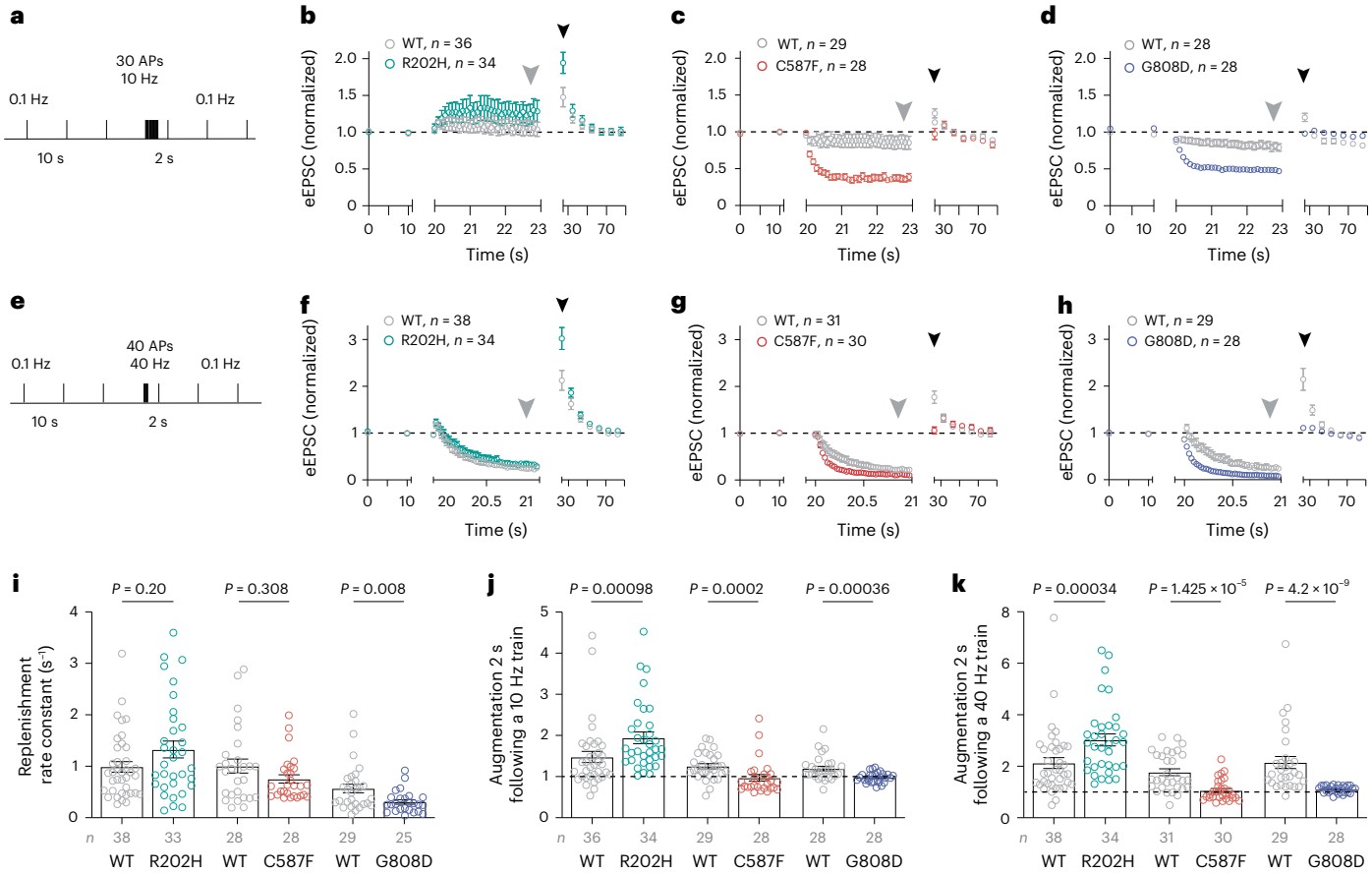

**Fig. 6 | UNC13A disease variants modulate neuronal short-term synaptic plasticity. a–d**, Stimulation protocol (**a**) and average time courses of normalized eEPSC amplitudes recorded in autaptic hippocampal neurons expressing UNC13A^WT (gray) or UNC13A^R202H (**b**, green), UNC13A^C587F (**c**, red) or UNC13A^G808D (**d**, blue), before, during and after a 10 Hz train consisting of 30 APs. **e–h**, Average time courses of normalized eEPSC amplitudes recorded before, during and after a 40 Hz train consisting of 40 APs (**e**) in neurons expressing UNC13A^WT (gray) or UNC13A^R202H (**f**, green), UNC13A^C587F (**g**, red), or UNC13A^G808D (**h**, blue). Gray arrowheads mark the time point during which steady-state depression was

quantified, and black arrowheads mark the first eEPSC following the train, used to estimate the magnitude of augmentation. **i**, Replenishment rate constants calculated based on a single-pool model of SV priming for neurons of the indicated genotypes. **j,k**, Magnitude of eEPSC augmentation after conditioning 10 Hz (**j**) or 40 Hz (**k**) stimulation. Data were obtained from at least three independent cultures per condition. Statistical analysis was performed by using a two-tailed Mann–Whitney test (**i–k**). Bars and circles in **i–k** represent means and values for individual neurons, respectively; error bars, s.e.m.

of milliseconds to minutes. STP occurs when fusion-competent SVs are progressively consumed. UNC13A-mediated SV priming and its modulation by second messengers counteracts SV depletion[10–13], thus shaping STP. We next studied how disease variants in UNC13A affect this process.

We recorded eEPSCs during high-frequency trains, embedded in low-frequency stimulation (Fig. 6a,e), and observed strong changes during and after stimulus trains (Fig. 6b–d,f–h): in neurons expressing UNC13A^C587F or UNC13A^G808D, prominent short-term synaptic depression of eEPSC amplitudes was observed (Fig. 6c,d,g,h (gray arrow)), whereas UNC13A^R202H-expressing neurons typically exhibited short-term synaptic enhancement (Fig. 6b and Extended Data Fig. 8). To explore underlying mechanisms, we quantified the replenishment rate constants of the RRP during 40 Hz AP trains, based on a single-pool model of SV priming[12,58]. RRP replenishment rate constants were reduced for neurons expressing UNC13A^C587F or UNC13A^G808D but tended to be accelerated in UNC13A^R202H-expressing neurons (Fig. 6i).

Following AP trains, synaptic strength can transiently increase (Fig. 6b–d,f–h (black arrowheads)). We quantified such augmentation by measuring eEPSCs 2 s after the cessation of AP trains. Compared to UNC13A^WT-expressing neurons, augmentation was significantly stronger when UNC13A^R202H was expressed, whereas it was virtually absent in neurons expressing UNC13A^C587F or UNC13A^G808D (Fig. 6j,k and Extended Data Figs. 5 and 8).

Collectively, these data demonstrate that UNC13A disease variants have a complex impact on STP. Reduced UNC13A expression favors facilitation of synaptic responses during and strong augmentation after an AP train, while gain of UNC13A function or diminished UNC13A PDBu/DAG regulation leads to increased depression during and reduced augmentation after AP trains.

## Discussion

We report the identification of a neurodevelopmental syndrome caused by pathogenic *UNC13A* variants that cause cell-autonomous changes to synaptic transmission and plasticity. The UNC13A syndrome joins the group of SNAREopathies that arise from pathogenic variants in the core proteins mediating neurotransmitter release (Extended Data Fig. 5a,b). The most common SNAREopathy is the *STXBP1* encephalopathy, which is diagnosed in 3.3–3.8 out of every 10,000 births[59,60]. *STXBP1* encodes the MUNC18-1 protein, a functional interactor of UNC13A in SNARE complex assembly[61–63].

Based on the genetic, clinical and functional evidence presented here (Extended Data Fig. 5c), we propose a categorization of the UNC13A syndrome. Type A comprises variants that cause full or partial UNC13A protein loss. This type is caused by biallelic missense (Fig. 3), splice-site (Extended Data Figs. 2 and 3) and nonsense variants based on our cohort and a few previously published cases[22,23]. These variants

lead to profound GDD in all cases, and in some, cause death in early childhood owing to respiratory failure after pneumonia (Fig. 1 and Supplementary Data 1). All patients develop early-onset seizures that mostly respond to treatment. Based on a previously reported case[23] in which an individual was diagnosed with fatal myasthenia according to electromyography findings, and based on patients 3 and 7 in our study (Supplementary Data 1 and Extended Data Figs. 2 and 3), we suggest considering myasthenic presentation in type A patients. Type A variants are associated with eliminated or strongly reduced UNC13A expression, weaker synaptic transmission under basal and low-frequency activity (Fig. 3) and a tendency towards enhanced synaptic responses during and after AP trains, or by DAG signaling (Figs. 5 and 6). We speculate that these variants impair UNC13A transport and/or active zone recruitment.

The type B UNC13A condition comprises variants that cause an UNC13A gain-of-function. This type is caused by de novo heterozygous missense variants in the UNC13A hinge (G808D, G808C, G808V, K811E, P814L) (Figs. 1 and 4) and is characterized by developmental delay, primarily in speech acquisition and, to a lesser extent, in motor skills. The overall severity of the developmental delay is less than that observed in type A, and no death in early childhood is reported. Epileptic seizures are severe and become refractory to treatment in most cases. A hallmark of this condition is ataxia and tremor or dyskinetic movements[24] (Fig. 1e, Supplementary Videos 1 and 2). As UNC13A is also expressed in motoneurons, it remains to be determined whether the imbalanced synaptic transmission at the neuromuscular junction contributes to this presentation. Functionally, type B variants cause a gain-of-function of synaptic transmission during spontaneous and low-frequency AP activity (Fig. 4), strong short-term synaptic depression and diminished augmentation during and after high-frequency activity (Fig. 6), as well as diminished potentiation by DAG signaling (Fig. 5).

Finally, the type C condition comprises a variant that results in UNC13A dysregulation. This type is caused by a familial heterozygous missense variant in the UNC13A C1 domain (C587F; Figs. 1 and 5), presenting with mild delay in speech development, learning difficulties to mild–moderate intellectual disability and controlled seizures (Supplementary Data 1). Neurons expressing this variant show near-normal synaptic transmission in response to low-frequency AP activity, but absent potentiation by DAG signaling (Fig. 5) and aberrant STP patterns during and after high-frequency activity (Fig. 6).

We anticipate additional pathogenic variants and disease subtypes to emerge. In particular, loss of function without loss of protein is expected for variants that interfere with UNC13A function in SNARE complex assembly (for example, in the MUN or C2C domain), which may lead to a type A-like condition. Based on AlphaMissense scores and the rareness of variants in the hinge region (frequency in gnomADv4.1 database), almost all possible hinge amino acid exchanges (42 out of 45; 93%) show pathogenic predictions, and we anticipate additional variants to arise. UNC13A hyperfunction has been described in structure–function studies[46,53,64], and variants in these regions may also lead to a type B condition. Missense or in-frame variants affecting $Ca^{2+}$ or $Ca^{2+}$/calmodulin binding may cause a type C condition.

Our functional assays were designed to investigate deficits caused by UNC13A variants and were therefore performed in the absence of a WT UNC13A allele (which is relevant for heterozygous variants) and in the absence of UNC13B/C. Co-expression of UNC13A/B/C is expected to modify the functional phenotypes we describe in a synapse-specific fashion, but to what extent is hard to predict and requires further experimentation. UNC13B and UNC13C are expressed alongside UNC13A in distinct neuronal subtypes of the cerebellum (UNC13C) and of the cortex and hippocampus (UNC13B)[17]. Interestingly, UNC13B expression can fully compensate for UNC13A loss in hippocampal inhibitory neurons, but UNC13B/C can only marginally compensate for UNC13A loss (<10%) in excitatory hippocampal synapses[20,65]. Reasons for the limited redundancy remain unclear and may involve active zone recruitment

mechanisms[66,67]. Nonetheless, following full or partial loss of UNC13A (type A), co-expression of UNC13B may support synaptic transmission in inhibitory neurons to a larger extent than in excitatory neurons, shaping the inhibition-to-excitation ratio within neuronal networks. It is tempting to speculate that this might underlie the favorable response of epileptic seizures to medication in type A patients. In the type B condition, we demonstrated that UNC13A[P814L] exerts a similar effect in excitatory hippocampal and inhibitory striatal neurons[24] and that the magnitude of the functional gain is similar in the presence or absence of UNC13A[WT] (Fig. 4n,o).

A key finding in this study is the identification of variants that reduced UNC13A expression levels below 50% of WT levels (E52K, R202H and, probably, some splice-site variants). Reduction of UNC13A expression levels has been recently highlighted as a pathogenic mechanism in ALS and FTD. In healthy neurons, TDP-43 acts as a splicing regulator for UNC13A mRNA[34,35,68], in which it represses the inclusion of a cryptic exon. In neurons with ALS or FTD pathology, TDP-43 is depleted from the nucleus[69], resulting in the inclusion of the cryptic exon in up to 100% of the UNC13A mRNA, and thus in lower UNC13A protein levels[34,35]. Deep intronic *UNC13A* variants that possibly change the cryptic exon's splicing propensity pose a strong genetic risk for ALS/FTD disease and disease progression[25,27,34,35]. How reduced UNC13A levels enhance ALS/FTD pathology remains to be established, and the minimal UNC13A levels required for proper synaptic function have not yet been determined. Several observations made here shed light on these issues.

We report that the R202H substitution severely interferes with human motor function (Fig. 1e), leads to reduced UNC13A expression levels (20–30% of WT levels; Fig. 2) and impairs synaptic function (Figs. 3, 5 and 6, and see ref. 70). Although we cannot exclude that R202H has additional effects on UNC13A function in addition to reducing its expression levels, these data suggest that altered neurotransmission may accompany ALS/FTD cellular pathology as UNC13A levels decline and potentially exacerbate disease symptoms. However, we also note that 20–30% of UNC13A expression levels are already sufficient to support ~60% of neurotransmitter release in cultured neurons and that patients with low UNC13A expression levels have improved survival chances compared with patients with no functional UNC13A, who die in early childhood[22,23]. Together, we propose that therapeutic strategies that stabilize 20–30% as minimal UNC13A expression may already be beneficial. Given that UNC13A haploinsufficiency appears to be tolerated in humans (refs. 22,23 and this study) and in mice[19], restoration of UNC13A to levels approaching 50% appears to be sufficient as an upper target. Antisense oligonucleotide therapies[71], for example, are approved for clinical use and could be well-used in ALS/FTD and in patients with type B and C neurodevelopmental disorders. In the latter, antisense oligonucleotides that exclusively suppress the expression of pathogenic transcripts and increase the relative abundance of the WT protein variant may improve disease symptoms. Our study underscores the importance of an experimentally validated mechanistic understanding of pathogenicity for diverse variants in the same gene, as different therapeutic strategies should be used to combat them.

## Online content

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

Reza Asadollahi [1,2] ✉, Aisha Ahmad[3,4], Paranchai Boonsawat[1], Jasmine Shahanoor Hinzen[5], Mareike Lohse [3,4], Boris Bouazza-Arostegui [6], Siqi Sun [4], Tillmann Utesch [4], Jonas D. Sommer[4], Dragana Ilic[4], Murugesh Padmanarayana[5], Kati Fischermanns[4], Mrinalini Ranjan[3], Moritz Boll[4], Chandran Ka[7,8], Amélie Piton [9], Francesca Mattioli[9], Bertrand Isidor[10], Katrin Õunap [11,12], Karit Reinson[11,12], Monica H. Wojcik [13,14,15], Christian R. Marshall [16], Saadet Mercimek-Andrews[17], Naomichi Matsumoto [18], Noriko Miyake [18,19], Bruno de Oliveira Stephan[20], Rachel Sayuri Honjo[20], Debora R. Bertola[20], Chong Ae Kim [20], Roman Yusupov[21], Heather C. Mefford [22], John Christodoulou [23,24], Joy Lee [25,26], Oliver Heath [26], Natasha J. Brown[24,26], Naomi Baker[26], Zornitza Stark [24,26], Martin Delatycki[26], Nicole J. Lake [23,27], Shimriet Zeidler [28], Linda Zuurbier[29], Saskia M. Maas[29], Chris C. de Kruiff [30], Farrah Rajabi[31,32], Lance H. Rodan[31], Stephanie A. Coury[31,33], Konrad Platzer [34], Henry Oppermann[34], Rami Abou Jamra [34], Skadi Beblo[35], Caroline Maxton[36], Robert Śmigiel [37], Hunter Underhill[38],

Holly Dubbs[39], Alyssa Rosen[39], Katherine L. Helbig[39,40], Ingo Helbig[39,40,41,42,43], Sarah McKeown Ruggiero[39,40], Mark P. Fitzgerald[39,40,43], Dennis Kraemer[1], Carlos E. Prada[44,45], Jeffrey Tenney[46,47], Parul Jayakar[48], Sylvia Redon[7,8], Jérémie Lefranc[49], Kevin Uguen[7,8], Simone Race[50], Stephanie Efthymiou[51], Reza Maroofian[51], Henry Houlden[51], Sandra Coppens[52], Nicolas Deconinck[53], Balasubramaniem Ashokkumar[54], Perumal Varalakshmi[54], Vykunta Raju Gowda K[55,56], Fatemeh Eghbal[57], Ehsan Ghayoor Karimiani[57], Morteza Heidari[58], John Neidhardt[59,60], Marta Owczarek-Lipska[59,60], G. Christoph Korenke[61], Michael J. Bamshad[62], Philippe M. Campeau[63], Anna Lehman[64], Laura G. Hendon[65], Ingrid M. Wentzensen[66], Kristin G. Monaghan[66], Yanmin Chen[66], Anna Szuto[67], Ronald D. Cohn[67], Ping Yee Billie Au[68], Christoph Hübner[69], Felix Boschann[70,71], Kandamurugu Manickam[72,73], Daniel C. Koboldt[73,74], Aboulfazl Rad[75], Gabriela Oprea[75], Kristine K. Bachman[76], Andrea H. Seeley[76], Emanuele Agolini[77], Alessandra Terracciano[77], Piscopo Carmelo[78], Caleb Bupp[79], Bethany Grysko[79], Annick Rein-Rothschild[80,81], Bruria Ben Zeev[81,82], Amy Margolin[83], Jennifer Morrison[83], Aditi Dagli[83], Elliot Stolerman[84], Raymond J. Louie[84], Camerun Washington[84], Servi J. C. Stevens[85], Malou Heijligers[85], Fowzan S. Alkuraya[86,87], Jasmin Lisfeld[88], Axel Neu[89], Fabíola Paoli Monteiro[90], André Luiz Santos Pessoa[91], Antonio Edvan Camelo-Filho[91], Fernando Kok[92], Dwight Koeberl[93], Kacie Riley[93], Lydie Burglen[94], Diane Doummar[95], Bénédicte Héron[95], Cyril Mignot[96], Boris Keren[96], Perrine Charles[96], Caroline Nava[96], Felix P. Bernhard[97], Andrea A. Kühn[98], Sven Thoms[99], Ryan D. Morrie[100], Shila Mekhoubad[100], Eric M. Green[100], Sami J. Barmada[101], Aaron D. Gitler[102], Olaf Jahn[103,104], Jeong Seop Rhee[3], Christian Rosenmund[6], Mišo Mitkovski[105], Heinrich Sticht[106], Han Sun[4], Gerald Le Gac[7,8], Holger Taschenberger[3], Nils Brose[3], Jeremy S. Dittman[5], Anita Rauch[1,107] & Noa Lipstein[3,4] ✉

[1]Institute of Medical Genetics, University of Zurich, Zurich, Switzerland. [2]Faculty of Engineering and Science, University of Greenwich London, Medway Campus, Chatham Maritime, London, UK. [3]Department of Molecular Neurobiology, Max Planck Institute for Multidisciplinary Sciences, Göttingen, Germany. [4]Leibniz-Forschungsinstitut für Molekulare Pharmakologie (FMP), Berlin, Germany. [5]Department of Biochemistry, Weill Cornell Medicine, New York, NY, USA. [6]Institute of Neurophysiology, Charité–Universitätsmedizin Berlin, corporate member of Freie Universität Berlin and Humboldt-Universität zu Berlin, Berlin, Germany. [7]Univ Brest, Inserm, EFS, UMR 1078, GGB, Brest, France. [8]Service de Génétique Médicale et Biologie de la Reproduction, CHU de Brest, Brest, France. [9]Institut de Génétique et de Biologie Moléculaire et Cellulaire, Strasbourg, France. [10]Nantes Université, CHU de Nantes, Service de Génétique médicale, Nantes, France. [11]Genetics and Personalized Medicine Clinic, Tartu University Hospital, Tartu, Estonia. [12]Institute of Clinical Medicine, University of Tartu, Tartu, Estonia. [13]Divisions of Newborn Medicine and Genetics and Genomics, Department of Pediatrics, Boston Children's Hospital, Harvard Medical School, Boston, MA, USA. [14]Manton Center for Orphan Disease Research, Division of Genetics and Genomics, Department of Pediatrics, Boston Children's Hospital, Harvard Medical School, Boston, MA, USA. [15]Broad Center for Mendelian Genomics, Broad Institute of MIT and Harvard, Cambridge, MA, USA. [16]Genome Diagnostics, Department of Paediatric Laboratory Medicine, The Hospital for Sick Children and University of Toronto, Toronto, Ontario, Canada. [17]Department of Medical Genetics, Faculty of Medicine & Dentistry, University of Alberta, Edmonton, Alberta, Canada. [18]Department of Human Genetics, Yokohama City University Graduate School of Medicine, Yokohama, Japan. [19]Department of Human Genetics, Research Institute, National Center for Global Health and Medicine, Tokyo, Japan. [20]Clinical Genetics Unit, Instituto da Crianca, Hospital das Clinicas HCFMUSP, Faculdade de Medicina, Universidade de São Paulo, São Paulo, Brazil. [21]Division of Clinical Genetics, Joe DiMaggio Children's Hospital, Hollywood, FL, USA. [22]Center for Pediatric Neurological Disease Research, St. Jude Children's Hospital, Memphis, TN, USA. [23]Brain and Mitochondrial Research Group, Murdoch Children's Research Institute, Melbourne, Victoria, Australia. [24]Department of Paediatrics, University of Melbourne, Melbourne, Victoria, Australia. [25]Department of Metabolic Medicine, The Royal Children's Hospital, Melbourne, Victoria, Australia. [26]Victorian Clinical Genetics Services, Murdoch Children's Research Institute, Melbourne, Victoria, Australia. [27]Department of Genetics, Yale School of Medicine, New Haven, CT, USA. [28]Department of Clinical Genetics, Erasmus MC, Rotterdam, The Netherlands. [29]Amsterdam UMC, University of Amsterdam, Department of Human Genetics, Amsterdam, The Netherlands. [30]Emma Children's Hospital, Amsterdam University Medical Centre, Amsterdam, The Netherlands. [31]Division of Genetics and Genomics, Boston Children's Hospital, Boston, MA, USA. [32]Department of Pediatrics, Section of Genetics and Metabolism, University of Colorado Anschutz Medical Campus, Aurora, CO, USA. [33]Division of Genetics, Brigham and Women's Hospital, Boston, MA, USA. [34]Institute of Human Genetics, University of Leipzig Medical Center, Leipzig, Germany. [35]Center for Pediatric Research, University Hospital for Children and Adolescents, and Centre for Rare Diseases, University Hospital Leipzig, Leipzig, Germany. [36]Zentrum für Kinderneurologie, Hamburg, Germany. [37]Department of Pediatrics, Endocrinology, Diabetology and Metabolic Diseases, Medical University of Wrocław, Wrocław, Poland. [38]Department of Pediatrics, Division of Medical Genetics, University of Utah, Salt Lake City, UT, USA. [39]Division of Neurology, Department of Pediatrics, Children's Hospital of Philadelphia, Philadelphia, PA, USA. [40]Epilepsy NeuroGenetics Initiative (ENGIN), Children's Hospital of Philadelphia, Philadelphia, PA, USA. [41]The Center for Epilepsy and Neurodevelopmental Disorders, Children's Hospital of Philadelphia, Philadelphia, PA, USA. [42]Department of Biomedical and Health Informatics (DBHi), Children's Hospital of Philadelphia, Philadelphia, PA, USA. [43]Department of Neurology, Perelman School of Medicine, University of Pennsylvania, Philadelphia, PA, USA. [44]Division of Genetics and Rare Diseases, Ann and Robert H. Lurie Children's Hospital of Chicago, Chicago, IL, USA. [45]Department of Pediatrics, Northwestern University Feinberg School of Medicine, Chicago, IL, USA. [46]Department of Pediatrics, University of Cincinnati College of Medicine, Cincinnati, OH, USA. [47]Division of Neurology, Cincinnati Children's Hospital Medical Center, Cincinnati, OH, USA. [48]Division of Genetics and Metabolism, Nicklaus Children's Hospital, Miami, FL, USA. [49]Pediatrics Department, Competence Center for Epilepsy, Hôpital Morvan, CHU Brest, Brest, France. [50]Division of Biochemical Genetics, BC Children's Hospital, Vancouver, British Columbia, Canada. [51]Department of Neuromuscular Disorders, UCL Queen Square Institute of Neurology, London, UK. [52]Center of Human Genetics, Hôpital Universitaire de Bruxelles, Université Libre de Bruxelles, Brussels, Belgium. [53]Centre de Référence Neuromusculaire and Paediatric Neurology Department, Hôpital Universitaire des Enfants Reine Fabiola, Hôpital Universitaire de Bruxelles (HUB), Université Libre de Bruxelles, Brussels, Belgium. [54]School of Biotechnology, Madurai Kamaraj University, Madurai, India. [55]Indira Gandhi Institute of Child Health, Bangalore, India. [56]Bangalore Child Neurology and Rehabilitation Center, Bangalore, India. [57]Department of Medical Genetics, Next Generation Genetic Polyclinic, Mashhad, Iran. [58]Myelin Disorders Clinic, Department of Pediatric Neurology, Children's Medical Center, Pediatrics Center of Excellence, Tehran University of Medical Sciences, Tehran, Iran. [59]Human Genetics, Medical Faculty, School of Medicine and Health Sciences, Carl von Ossietzky Universität Oldenburg, Oldenburg, Germany. [60]Research Center Neurosensory Science, Carl von Ossietzky

University Oldenburg, Oldenburg, Germany. [61]Department of Neuropediatrics, University Children's Hospital, Klinikum Oldenburg, Oldenburg, Germany. [62]Division of Genetic Medicine, Department of Pediatrics, University of Washington, Seattle, WA, USA. [63]Department of Pediatrics, CHU Sainte-Justine and University of Montreal, Montreal, Quebec, Canada. [64]Department of Medical Genetics, The University of British Columbia, Vancouver, British Columbia, Canada. [65]Department of Pediatrics, University of Mississippi Medical Center, Jackson, MS, USA. [66]GeneDx, LLC, Gaithersburg, MD, USA. [67]The Hospital for Sick Children, Toronto, Ontario, Canada. [68]Department of Medical Genetics, Alberta Children's Hospital Research Institute, Cumming School of Medicine, University of Calgary, Calgary, Alberta, Canada. [69]Department of Neuropediatrics, Faculty of Medicine and University Hospital Carl Gustav Carus, Technische Universität Dresden, Dresden, Germany. [70]Institut für Medizinische Genetik und Humangenetik, Charité–Universitätsmedizin Berlin, corporate member of Freie Universität Berlin and Humboldt-Universität zu Berlin, Berlin, Germany. [71]BIH Biomedical Innovation Academy, Clinician Scientist Program, Berlin Institute of Health at Charité–Universitätsmedizin Berlin, Berlin, Germany. [72]Division of Genetic and Genomic Medicine, Nationwide Children's Hospital, Columbus, OH, USA. [73]Department of Pediatrics, The Ohio State University College of Medicine, Columbus, OH, USA. [74]The Steve and Cindy Rasmussen Institute for Genomic Medicine, Nationwide Children's Hospital, Columbus, OH, USA. [75]Arcensus GmbH, Rostock, Germany. [76]Department of Pediatrics, Women's and Children's Institute, Geisinger Medical Center, Danville, PA, USA. [77]Laboratory of Medical Genetics, Translational Cytogenomics Research Unit, Bambino Gesù Children's Hospital, IRCCS, Rome, Italy. [78]Medical and Laboratory Genetics Unit, A.O.R.N. "Antonio Cardarelli", Naples, Italy. [79]Corewell Health West Helen DeVos Children's Hospital, Grand Rapids, MI, USA. [80]The Institute for Rare Diseases, Edmond and Lily Safra Children's Hospital, Sheba Medical Center, Ramat Gan, Israel. [81]School of Medicine, Faculty of Medical and Health Sciences, Tel Aviv University, Tel Aviv, Israel. [82]Pediatric Neurology Unit, The Edmond and Lily Safra Children's Hospital, Sheba Medical Center, Ramat Gan, Israel. [83]Division of Medical Genetics, Arnold Palmer Hospital Orlando Health, Orlando, FL, USA. [84]Greenwood Genetic Center, Greenwood, SC, USA. [85]Department of Clinical Genetics, Maastricht University Medical Center, Maastricht, The Netherlands. [86]Department of Translational Genomics, Center for Genomic Medicine, King Faisal Specialist Hospital and Research Center, Riyadh, Saudi Arabia. [87]College of Medicine, Alfaisal University, Riyadh, Saudi Arabia. [88]Institute of Human Genetics, University Medical Center Hamburg-Eppendorf, Hamburg, Germany. [89]Department of Pediatrics, University Medical Center Hamburg-Eppendorf, Hamburg, Germany. [90]Medical Department, Mendelics Genomic Analysis, São Paulo, Brazil. [91]Federal University of Ceará - UFC and Hospital Infantil Albert Sabin, Fortaleza, Brazil. [92]Department of Neurology, Neurogenetics Center, University of São Paulo, São Paulo, Brazil. [93]Division of Medical Genetics, Department of Pediatrics, Duke University School of Medicine, Durham, NC, USA. [94]Département de Génétique, Centre de référence des malformations et maladies congénitales du cervelet, Hôpital Trousseau, AP-HP.Sorbonne Université, Paris, France. [95]Service de Neuropédiatrie-Pathologie du Développement, Centre de Référence Neurogénétique, Hôpital Trousseau, FHU I2-D2, AP-HP.Sorbonne Université, Paris, France. [96]Département de génétique, Hôpital Pitié-Salpêtrière, Centre de Références Déficiences Intellectuelles de Causes Rares, AP-HP.Sorbonne Université, Paris, France. [97]Department of Psychiatry and Psychotherapy, University of Marburg, Marburg, Germany. [98]Movement Disorders and Neuromodulation Unit, Department of Neurology, Charité–Universitätsmedizin Berlin, Berlin, Germany. [99]Biochemistry and Molecular Medicine, Medical School OWL, Bielefeld University, Bielefeld, Germany. [100]Trace Neuroscience, South San Francisco, CA, USA. [101]University of Michigan School of Medicine, Ann Arbor, MI, USA. [102]Department of Genetics, Stanford University School of Medicine, Stanford, CA, USA. [103]Neuroproteomics Group, Department of Molecular Neurobiology, Max Planck Institute for Multidisciplinary Sciences, Göttingen, Germany. [104]Translational Neuroproteomics Group, Department of Psychiatry and Psychotherapy, University Medical Center Göttingen, Göttingen, Germany. [105]City Campus Light Microscopy Facility, Max Planck Institute for Multidisciplinary Sciences, Göttingen, Germany. [106]Institute of Biochemistry, Friedrich-Alexander-Universität Erlangen-Nürnberg (FAU), Erlangen, Germany. [107]Children's Hospital, University Zurich, Zurich, Switzerland. ✉e-mail: R.Asadollahi@greenwich.ac.uk; Lipstein@FMP-berlin.de

## Methods

### Study approval

This study was performed as part of a research study approved by the ethics commission of the Canton of Zurich (ID PB_2016-02520 (SIV 11/09)). In addition, for each patient, ethical approvals and informed consent forms from parents or guardians were obtained by the respective research teams and institutions for data use and publication, including photographs or videos where applicable. Participants did not receive compensation. The use of Unc13a/b knockout mice was approved by the responsible local government organization (Niedersächsisches Landesamt für Verbraucherschutz und Lebensmittelsicherheit; 33.19-42502-04-15/1817 and 33.19-42502-04-20/3589, and Landesamt für Gesundheit und Soziales; G106/20). Mice (*Mus musculus*, backcrossed to a C57BL/6N background) of embryonic day 18 or postnatal day 0–1 and of both sexes were used in electrophysiological experiments. Each culture was obtained from one or more mice, and two to four independent cultures were used per condition. Adult mice were kept under IVC/SPF conditions, on a 12 h light, 12 h dark cycle, at room temperature ($22 \pm 2$ °C) and humidity levels of $55 \pm 10\%$.

### Patient identification and characterization

Individuals included in this study underwent exome or genome sequencing on diverse sequencing platforms on a clinical or research basis in academic institutes or diagnostic labs worldwide. Using GeneMatcher[73] and personal communication with colleagues enabled us to assemble clinical and genetic details on a total of 48 index patients with de novo or inherited heterozygous or biallelic variants in *UNC13A* (NM_001080421.3), including two previously published cases harboring c.154 G > A, p.(E52K) (case 74 described in ref. [74]) and c.4379 C > T, p.(A1460V) (case UPN-0740 described in ref. [75]) variants with further follow-up information. Standard deviation of growth parameters was calculated as previously described[76]. Variant nomenclature was verified using VariantValidator[77]. De-identified clinical data from collaborating institutions were obtained with local institutional review board approval and informed consent from parents or guardians of affected individuals, including consent for publication of facial photographs for individuals in Figs. 3–5, Extended Data Figs. 1 and 2 and Supplementary Videos 1 and 2.

### Minigene splicing assay

For splicing assays, the pSplice*POLR2G3* vector was digested and purified as described previously[78]. Inserts of *UNC13A* exons and their flanking intronic regions were amplified using the Q5 HotStar High-Fidelity Taq Polymerase kit (New England Biolabs), and cloning reactions were performed with the In-Fusion HD Cloning kit (Clontech) following the manufacturer's recommendations. Transient transfection of HEK293T cells was performed with WT and mutated plasmid constructs (1 µg per well) complexed with the jetPEI™ transfection reagent (Polyplus Transfection). Total RNA was extracted 48 h post transfection with the NucleoSpin RNA Plus kit (Macherey-Nagel). Reverse transcription was performed with the ProtoScript II First Strand cDNA Synthesis Kit (New England Biolabs), and fluorescent PCR was performed as described previously[79]. PCR products were size-separated on a 3130xl Genetic Analyzer (Applied Biosystems). Output data were analyzed with Peak Scanner software (Applied Biosystems).

### DNA construct and virus preparation

Unless otherwise indicated, cDNA encoding rat UNC13A tagged with EGFP at the carboxy terminus[80] was used. Mutated amino acids are indicated throughout the paper according to the human nomenclature, and Supplementary Table 1 lists the respective amino acids mutated in the rat cDNA. Lentiviral preparation was carried out by the Viral Core Facility of the Charité–Universitätsmedizin Berlin (vcf.charite.de). The FUGW vector[81], in which the ubiquitin promoter was exchanged with the human synapsin 1 promoter, was used according to a previously published[81], modified protocol[82], using helper plasmids (Addgene, 8454 and 8455; ref. [44]). In a subset of experiments, constructs encoding NLS-GFP-P2A-UNC13A-Flag under the synapsin promoter were used[82]. In Fig. 5e, pEGFP–N1–UNC13A^WT and corresponding mutants were used. Viral transfection of neurons was carried out on DIV 2–3. Plasmid and primer sequence information are summarized in Supplementary Table 1.

### Antibodies

Antibodies used in this study are listed in Supplementary Table 2.

### Microisland cultures of mouse hippocampal neurons and electrophysiological recordings

Neurons were prepared from the brains of E18 Unc13a/b DKO mice and plated on WT astrocyte microisland cultures according to published protocols[83]. Cultures were kept at 37 °C and 5% $CO_2$ until recordings were made at DIV 13–16. Whole-cell voltage-clamp data were acquired using an Axon Multiclamp 700B amplifier, Digidata 1440A data acquisition system and pCLAMP 10 software (Molecular Devices). All recordings were performed using a standard external solution containing 140 mM NaCl, 2.4 mM KCl, 10 mM HEPES, 10 mM glucose, 4 mM $CaCl_2$ and 4 mM $MgCl_2$ (320 mOsm l$^{-1}$). The standard internal solution contained 136 mM KCl, 17.8 mM HEPES, 1 mM EGTA, 4.6 mM $MgCl_2$, 4 mM NaATP, 0.3 mM $Na_2$GTP, 15 mM creatine phosphate and 5 U ml$^{-1}$ phosphocreatine kinase (315–320 mOsm l$^{-1}$), pH 7.4. eEPSCs were evoked by depolarizing the cell from −70 mV to 0 mV for a 1 ms duration. Basal eEPSCs were recorded at a frequency of 0.1 Hz. Rapid external solution exchange around individual neurons was achieved by using a custom-made fast-flow system controlled by a stepper motor. Recordings were made at room temperature (-22 °C). The size of the RRP was estimated from the integrated charge of EPSCs evoked by application of 500 mM sucrose (7 s or 5 s in Fig. 3k–l) after subtraction of the steady-state current to correct for ongoing SV pool replenishment. mEPSCs were recorded for 100 s in the presence of 300 nM tetrodotoxin. In Fig. 3g,h, additional recordings in the presence of NBQX (3 µM, Tocris Bioscience) were used to subtract false-positive events from traces recorded in extracellular solution. mEPSC traces were filtered at 1 kHz, and miniature events were identified using a sliding template function in Axograph or in Igor Pro (5–200 pA; rise time, 0.15–1.5 ms; half-width, 0.5–5 ms). PDBu (1 µM) was applied for 1 min. Analyses were performed using Axograph (v.1.4.3) (Molecular Devices) or Igor Pro (Wavemetrics). Data were obtained from at least two cultures, and for the majority of data, from four cultures.

### Immunocytochemistry

Quantifications of synapse number, signal intensity and colocalization were made by immunostaining of microisland hippocampal cultures. Lentiviral-transfected samples were briefly washed with PBS and fixed at DIV 13 with 4% paraformaldehyde (PFA) in PBS for 18 min with gentle agitation. After three PBS washes, the remaining PFA was quenched with 50 mM glycine for 10 min, and samples were washed again with PBS. Before permeabilization, samples were treated with four drops of Image-iT FX signal enhancer in 1.5–2 ml PBS per well (in a six-well plate) and maintained shaking for 20 min. Samples were permeabilized for 30 min with 0.1% Triton X-100 and 2.5% normal goat serum (NGS) diluted in PBS while gently shaking. Blocking was performed with 2.5% NGS in PBS, twice for a total of 15 min. The primary antibodies were diluted in blocking solution, and coverslips were mounted upside down on 150 µl antibody solution placed on parafilm and kept for 2.5 h at room temperature or overnight at 4 °C, both in a wet chamber. Coverslips were washed three times with 0.1% Triton X-100 and 2.5% NGS in PBS and subsequently mounted upside down in 150 µl secondary antibody solution in a dark, wet chamber for 45 min at room temperature. Finally, the coverslips were washed five times with 0.1% Triton X-100 and 2.5% NGS in PBS for 2–5 min. The coverslips were briefly washed in

double-distilled water, excess water was removed and the coverslips were mounted on a glass slide using Aqua-Poly/Mount.

## Image acquisition and analysis

Microislands containing one hippocampal neuron were efficiently localized by first acquiring a low-magnification overview image of the entire coverslip using a Zeiss LSM 880 confocal laser scanning microscope. Specifically, upon 405 nm excitation, a ×10 air objective (0.45 NA) was used to image Alexa Fluor 405-stained MAP2 (410–470 nm). The overview image served as a map based on which the motorized stage of the Zeiss LSM 880 was moved to selected autaptic neurons, from which high-resolution ($x$, $y$, $z$: 208, 208, 309 nm) $z$-stacks of VGLUT1 (637–758 nm), Shank2 (570–624 nm), UNC13A–eGFP (490–552 nm) and MAP2 (410–470 nm) were acquired with a ×40 oil immersion objective (1.40 NA). Overlapping fields of view for subsequent image stitching were acquired with the above settings, in case the autaptic neuron did not fit in one field of view.

All image processing was conducted with the same automated Imaris (v.9.10.0) batch process. Initially, the respective MAP2 three-dimensional surface was generated, followed by surfaces based on the respective VGLUT1, Shank2 or UNC13A–eGFP signal located within the 0.5 μm MAP2 periphery. Overlapping Shank2 and VGLUT1 surfaces were used to estimate the total number of synapses contacting individual autaptic neurons. Subsequently, the synapses overlapping with the UNC13A–eGFP signal were identified, and in a reciprocal manner, UNC13A–eGFP surfaces that overlap with synapses were identified. The data generated by the Imaris batch process were organized with the KNIME Analytics Platform (v.4.7.1) and further quantified with GraphPad Prism (v.9.0.0). For illustration, representative images were processed with the FIJI software package[84] (v.2.14.0/1.54f) and assembled in Adobe Illustrator.

## Translocation assay

HEK293T cells were seeded on poly-L-lysine-coated glass coverslips and cultured in DMEM supplemented with 1% MEM, 1% GlutaMAX, 1% penicillin–streptomycin and 10% FBS at 37 °C under 5% $CO_2$. Cells were transfected with UNC13A–GFP cDNA in pEGFP–N1 vector[80] carrying the respective disease-related variations under the CMV promoter using Lipofectamine 2000 and 1 μg DNA per well in a six-well plate, according to the manufacturer's recommendations. After 3.5 h of incubation, the medium was replaced with culture medium, and the cells were incubated overnight. Translocation was induced by applying 1 μM PDBu for 1 h. Cells were washed with PBS, fixed with 4% PFA in PBS at room temperature for 10 min, washed with PBS and treated with 50 mM glycine in PBS for 10 min to quench remaining PFA. Cells were washed in PBS and incubated with 1 μg ml$^{-1}$ DAPI in PBS at room temperature for 15 min. Cells were then washed again with PBS and mounted using Aqua-Poly/Mount. Fluorescence imaging was performed on a confocal laser scanning microscope (Zeiss LSM710-CONFOCOR3). Excitation wavelengths were 405 nm for DAPI and 488 nm for EGFP.

## *C. elegans* strains and behavioral assays

Strains were maintained and genetically manipulated as previously described[85]. Animals were raised at 20 °C on nematode growth media seeded with OP50. Control strains used included *N2* and *unc-13(nu641)*. *unc-13(nu641)* harbors a C-terminal mScarlet in the *unc-13* locus as described previously[86]. UNC-13 residue numbering is based on the UNC-13L isoform accession number NP_001021874. CRISPR–Cas9-mediated genome editing was performed to generate the desired genetic modifications in *C. elegans*. The CRISPR–Cas9 system was implemented using a co-injection strategy with Cas9 protein, CRISPR RNA (crRNAs) and repair templates.

For crRNA design, crRNAs targeting the desired genomic loci were initially designed using the Benchling online tool and custom crRNA design software from Integrated DNA Technologies (IDT). crRNAs were

selected based on proximity of the cut site to the desired mutation, high on-target efficiency and minimal predicted off-target mutations. crRNAs were ordered from IDT, resuspended in 4 μg μl$^{-1}$ in nuclease-free water and diluted to a working concentration of 0.4 μg μl$^{-1}$ before adding to the injection mix.

For Cas9 preparation, recombinant Cas9 protein purchased from IDT (Alt-R S.p. Cas9 Nuclease V3) was stored at −20 °C and used at 0.25 mg ml$^{-1}$. Trans-activating CRISPR RNA (tracrRNA) was purchased from IDT, diluted to 4 μg μl$^{-1}$ in IDT nuclease-free duplex buffer and aliquots were stored at −20 °C.

**Repair template design.** For precise edits, single-stranded oligodeoxynucleotides were used as repair templates. These templates included the desired modification flanked by 20–50 bp homology arms and an engineered restriction site. Single-stranded oligodeoxynucleotides were ordered from IDT (Ultramer DNA Oligonucleotides).

**Injection mix preparation.** The injection mix was prepared as follows: 0.5 μl Cas9 protein (10 mg ml$^{-1}$ stock), 5 μl tracrRNA (0.4 μg μl$^{-1}$ stock), 2.8 μl mutant crRNA (0.4 μg μl$^{-1}$ stock) and 2.8 μl mutant co-CRISPR crRNA (0.4 μg μl$^{-1}$ stock). The mixture was incubated at 37 °C for 15 min to allow ribonucleoprotein complex formation. If a PCR repair template was used, it was melted during this step. Following ribonucleoprotein formation, 2 μl of mutant repair oligonucleotide (100 μM stock) and 2 μl of co-CRISPR repair oligonucleotide were added. Nuclease-free water was added to bring the final volume to 20 μl.

**Microinjection and recovery.** Young adult worms (either *N2* or *unc-13(nu641)*) were immobilized on 2% agarose pads in halocarbon oil. The CRISPR–Cas9 injection mix was manually injected into the gonads of P0 worms using a Picospritzer III microinjector (Parker). The worm gonad was visualized using an Olympus IX51 inverted microscope with a ×40 air objective. Injected worms were recovered on seeded nematode growth medium plates at 20 °C for 3–4 days.

**Screening for edited worms.** F1 progeny expressing the co-injection marker (*unc-58*) were selected for screening and grown to the young adult stage. Genomic DNA was extracted from worm lysates, and the targeted locus was amplified by PCR. PCR products were digested with the engineered restriction site to identify potential edits. Worms containing the restriction site and lacking the *unc-58* gain-of-function phenotype were singled out, and homozygous progeny were isolated. Genomic DNA was then re-extracted and sequenced using nanopore sequencing (Plasmidsaurus) to confirm the presence of the desired edits. Animals were then outcrossed at least four times with the parent strain. Sequence information for *C. elegans* strains can be found in Supplementary Note 1.

**Aldicarb assay.** To measure aldicarb sensitivity in *C. elegans*, 20–25 animals were placed on OP50-seeded agar plates containing 1 mM aldicarb (CarboSynth) as described previously[87,88]. Animals were scored for paralysis every 10 min for 2 h for each genotype. During an assay, the experimenter was blind to all genotypes, and each genotype was assayed at least ten times. Paralysis curves for each genotype were generated by averaging individual time courses.

## Molecular model building and molecular dynamics simulations

A model of UNC13A consisting of the C1, C2B and MUN domains (residues 505–985) was predicted with AlphaFold (v.2.0; 01-JUL-21)[89] using the UniProt sequence Q9UPW8 as input. The resulting model showed, as expected, a high similarity (root mean squared deviation, 1.08 Å) to the known 3D structure of rat MUNC-13 (PDB 5UE8)[90]. In addition to the protein, the two $Zn^{2+}$ ions in the C1 domain were included. The $Zn^{2+}$-coordinating cysteines and histidines were modeled as unprotonated and protonated on their $N_\varepsilon$, respectively. All other amino acids

were protonated at pH 7.0. Single mutations in the loop region (G808C, G808D, K811E, P814L), adjacent to the loop R799Q, and at the C1 domain (C587F, H554K) were incorporated with PyMol. Subsequently, WT UNC13A and all single mutants were solvated in cubic boxes of SPC/E water[91] with the dimension of approximately 12.8 nm$^3$. The net charges of the models were neutralized with K$^+$Cl$^-$, and an ionic strength of 150 mM K$^+$Cl$^-$ was added. Additionally, five Ca$^{2+}$ ions were inserted, replacing water molecules. To prevent ion aggregation, a previously published parameter set[92] was applied. The protein was described by the Amber99SB*-ILDN force field[93].

The subsequent molecular dynamics simulations were performed with the Gromacs 2021.2 package. After energy minimization and thermal equilibration with gradually decreasing position restraints, all periodic models were allowed to move freely during the following 1 μs-long production runs. To ensure reasonable statistics, all simulations were repeated five times. The temperature was maintained at 300 K and the pressure at 1 atm with the V-rescale scheme[94] and the Parrinello-Rahman barostat[95], respectively.

Constraining all bonds containing hydrogen atoms with the Lincs algorithm[96] allowed a time step of 2 fs. Short-range electrostatics and van der Waals interactions were truncated at 1.2 nm, while long-range electrostatics were calculated with the Particle Ewald summation[97].

**Analysis.** Images of 3D structures were generated with VMD[98], graphs were drawn with the Matplotlib library included in Python[99], and dipole moments were calculated with the MDAnalysis tool[100].

### Protein stability assay

HEK293T cells were transfected with a plasmid encoding WT GFP–P2A–FLAG–MUNC13-1 without or with disease-related variants. As a negative control, non-transfected cells were used. Cells were transfected with Lipofectamine 2000 (according to the manufacturer's manual) and 1 μg of the plasmid. Then, 24 h after transfection, cells were lysed in RIPA buffer (150 mM NaCl, 1% IGEPAL CA 630, 1% sodium-deoxycholate, 0.1% SDS and 30 mM Tris-Cl, pH 7.4, supplemented with aprotinin, leupeptin and phenylmethylsulfonyl fluoride). Lysates were sonicated and centrifuged at 4 °C for 15 min at 21,000g. Supernatants were transferred into a fresh tube, and protein concentration was determined. Samples were diluted in Laemmli buffer and incubated for 5 min at 95 °C. Next, 10 μg of the samples were separated on 8% Bis–Tris gel in cold MOPS buffer at 150 V, blotted on a nitrocellulose membrane (Protran 0.2 NC, Amersham) and stained with the Pierce reversible protein stain and destain kit (according to the manufacturer's manual; Thermo Fisher Scientific, 46430) for a sample loading control. The membranes were cut and incubated with an anti-Flag antibody, anti-GFP antibody (see Supplementary Table 2) and corresponding secondary antibodies, and the chemiluminescence signal was detected on an ECL ChemoStar imager (Intas). Signal intensity quantification was done in Fiji[101]. Protein levels were first normalized to the total protein staining of the corresponding lane, and ratios between the GFP and Flag signals were calculated.

### Statistics

Electrophysiological data are presented as means ± s.e.m., and statistical significance was determined using the non-parametric two-tailed Mann–Whitney test. For the *C. elegans* paralysis assay, data were analyzed using a non-parametric Kruskal–Wallis test followed by Dunn's test for all pairwise comparisons. Western blots and imaging data are presented as means ± s.e.m., and statistical significance was evaluated using a non-parametric Kruskal–Wallis test. All analyses used two-sided tests and were performed using R statistical software (v.4.4.2)[102] and the R packages 'coin' and 'dunn.test'.

### Reporting summary

Further information on research design is available in the Nature Portfolio Reporting Summary linked to this article.

## Data availability

All data supporting the findings of this study are available within the paper, Supplementary Data 1–3, the Source Data file and in the Supplemental Information file. Molecular dynamics simulation data are available at https://doi.org/10.5281/zenodo.14554776 (ref. 103). All other primary data will be made available on request from the corresponding authors. Publicly available databases used in this study include GeneMatcher (https://genematcher.org), MetaDome web server (https://stuart.radboundumc.nl/metadome/), ClinVAR (https://www.ncbi.nlm.nih.gov/clinvar), Genome Aggregation Database (gnomAD; http://gnomad.broadinstitute.org), OMIM (http://www.omim.org), UCSC Genome Browser (https://genome.ucsc.edu), Ensembl (https://www.ensembl.org) and UniProt (https://www.uniprot.org). Source data are provided with this paper.

## Code availability

No code was developed for this study.

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

## Acknowledgements

We thank all involved families for their participation and permission to publish this study. We thank A. Günther, K. Steinhagen, the AGCT lab and animal facility staff at the Max Planck Institute of Multidisciplinary Sciences for technical help, and B. van Rossum for illustrations. We thank the Viral Core Facility, Charité–Universitätsmedizin Berlin, for virus production, and B. Weinberg for the contribution of lentiviral vectors (Addgene, 8454 and 8455). This work was supported by the German Research Foundation Excellence Strategy EXC-2049-390688087 (N.L., C.R., A.A.K.); CRC1286 (A11, N.L.; A9, N. Brose), EXC2067/1-390729940 (N. Brose), ERC Advanced Grant SynPrime (N. Brose), TargetALS (N.L., A.D.G., R.D.M., S.M., E.M.G., S.J.B.), the North-German Supercomputing Alliance HLRN (H. Sun and T.U.), the University of Zurich Clinical Research Priority Program Praeclare (A. Rauch), the University of Greenwich QR Funding (R.A.), NIH NS 116747 (to J.S.D.) and 1F31NS130894-01A1 (to J.S.H.). Additional acknowledgments are listed in Supplementary Note 2. The funders had no role in study design, data collection and analysis, decision to publish or preparation of the manuscript.

## Author contributions

N.L., R.A., A. Rauch, H.T. and N. Brose conceptualized the study. N.L., R.A., A. Rauch, H.T., C.R., J.S.R., H. Sun, M.M., J.S.D., G.L.G. and H. Sticht devised the methodology. N.L., A.A., B.B.-A., H.T., R.A., M.L., M.M., M.P., D.I., P.B., T.U., M.B., M.R., J.S.D., G.L.G. and S.T. conducted the formal analysis. N.L., A.A., R.A., P.B., C.K., J.S.H., M.L., B.B.-A., S.S., T.U., J.D.S., D.I., O.J., M.B., M.P., K.F., D. Kraemer, J.D.S. and H. Sticht performed the investigation. N.L., N. Brose and R.A. wrote the original draft of the paper. N.L., R.A., P.B., M.L., M.M., T.U., H. Sticht, B.B.-A., C.K., G.L.G., J.S.D. and A.E.C.-F. were responsible for visualization. N.L., R.A., A. Rauch, N. Brose, C.R., J.S.D. and G.L.G. supervised the study. N.L., R.A., A. Rauch and N. Brose acquired funding. A.A.K., A. Rad, A.D., A.D.G., A.E.C.-F., A.H.S., A.L., A. Rosen, A.L.S.P., A.M., A.N., A.P., A.R.-R., A.S., A.T., B.A., B.B.Z., B.d.O.S., B.G., B.H., B.I., B.K., C.A.K., C. Maxton, C.B., C.C.d.K., C.E.P., C.H., C.N., C.R.M., C.W., C. Mignot, D.C.K., D.D., D.R.B., D. Koeberl, E.A., E.G.K., E.M.G., E.S., F.B., F.E., F.K., F.M., F.P.B., F.P.M., F.R., F.S.A., G.C.K., G.O., H.C.M., H.D., H.H., H.O., H.U., I.H., I.M.W., J. Lisfeld, J.C., J. Lefranc, J.M., J.N., J. Lee, J.T., K. Riley, K. Reinson, K.G.M., K.L.H., K.K.B., K.M., K.Õ., K.P., K.U., L.B., L.G.H., L.H.R., L.Z., M. Heijligers, M.D., M.H.W., M.J.B., M. Heidari, M.O.-L., M.P.F., N. Baker, N. Matsumoto, N.D., N.J.B., N.J.L., N. Miyake, O.H., P. Charles, P. Carmelo, P.J., P.M.C., P.V., P.Y.B.A., R.A.J., R.C., R.M., R.S.H., R.J.L., R.Ś., R.Y., R.D.M., S.A.C., S.C., S.B., S.E., S. Race, S.J.B., S.J.C.S., S.M., S.M.-A., S.M.M., S.M.R., S. Redon, S.Z., V.R.G.K., Y.C. and Z.S. acquired resources (genetic, diagnostic and clinical information). All authors read, reviewed, edited and approved the paper.

## Funding

V. (FMP).

## Competing interests

A.D.G. is a scientific founder of Trace Neuroscience. N.L. is a member of the scientific advisory board of Trace Neuroscience. S.M., E.M.G. and R.D.M. are employees of Trace Neuroscience. All the above may own stock in Trace Neuroscience. G.O. and A. Rad are employees of Arcensus GmbH. Y.C., K.G.M. and I.M.W. are employees of and may own stock in GeneDx. D. Koeberl has received grant support from Viking Therapeutics, Genzyme Sanofi, Roivant Rare Diseases and Amicus and has held equity in Asklepios Biopharmaceutical (AskBio). The other authors declare no competing interests.

## Additional information

**Extended data** is available for this paper at https://doi.org/10.1038/s41588-025-02361-5.

**Correspondence and requests for materials** should be addressed to Reza Asadollahi or Noa Lipstein.

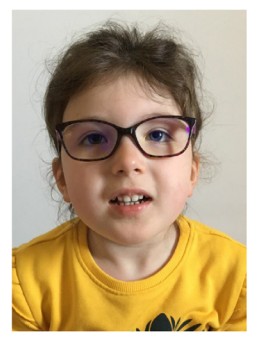
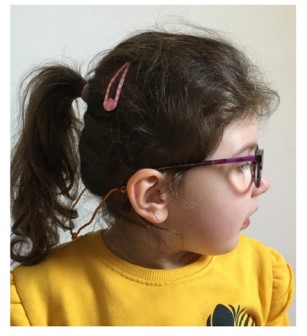
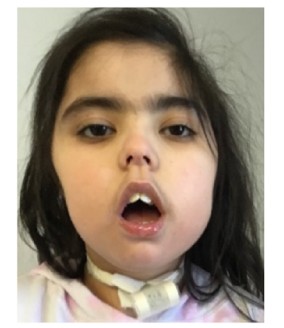
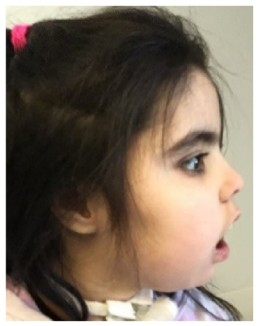

c.52+1G>A; p.(T22M), CH

p.(G484S); p.(R1495H), CH (10y6m)

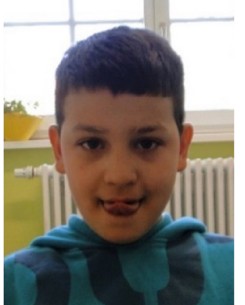
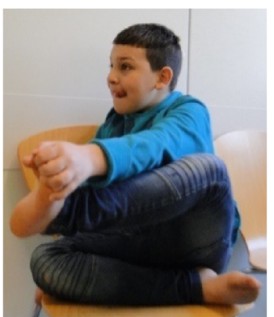
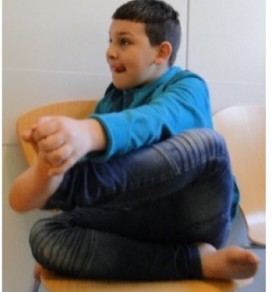
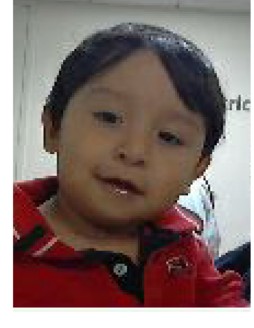
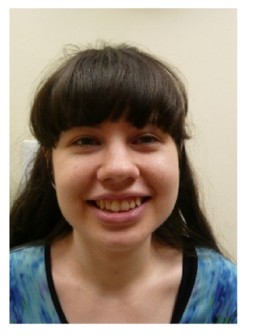
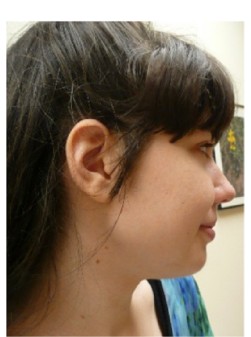

p.(G929E), DN

p.(S1083R), DN

p.(Q1349P), DN (20y)

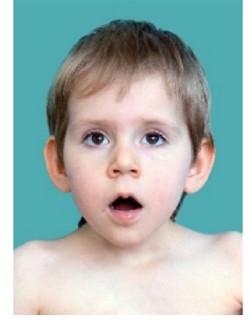
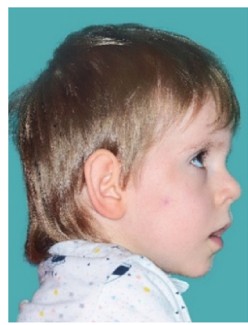
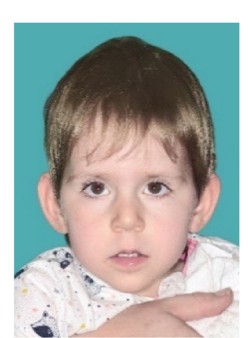

p.(P1596R); c.4811+2_4811+3delTG, CH

**Extended Data Fig. 1 | Facial photographs of six patients from this study harboring compound heterozygous (CH) or de novo (DN)** *UNC13A* **variants of uncertain significance (VUS).** See details in Supplementary Table 1.

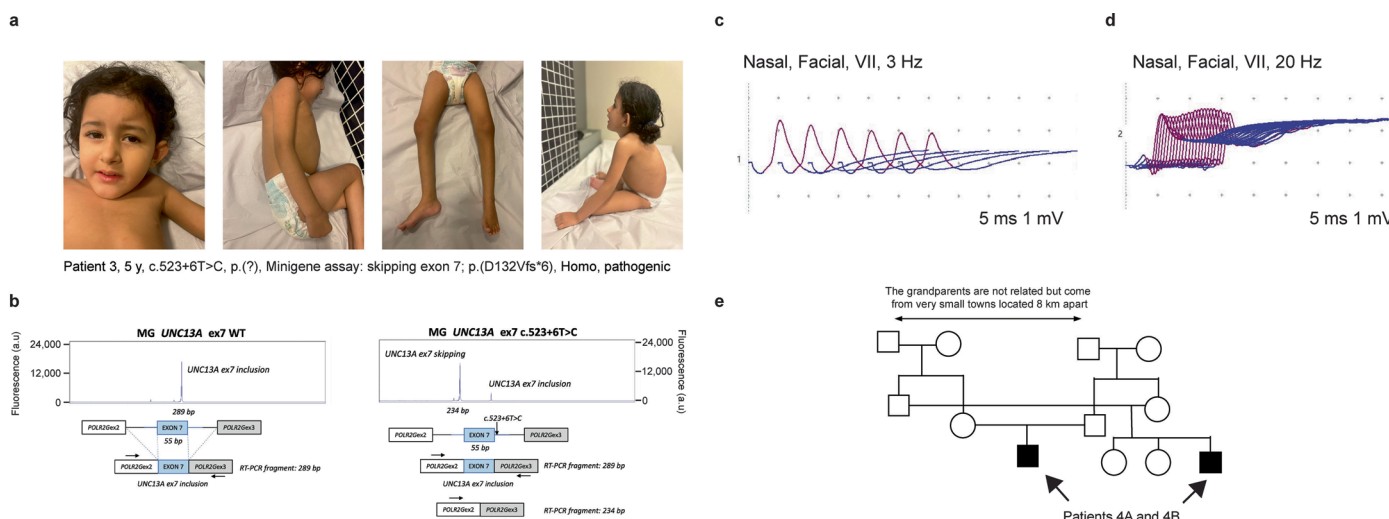

**Extended Data Fig. 2 | Clinical and/or genetic information for patients 3 and 4. a**, Photographs of patient 3 harboring a pathogenic homozygous variant c.523+6 T > C in intron 7 of *UNC13A*. **b**, *UNC13A* c.523+6 T > C variant exhibits a strong, deleterious splicing impact leading to exon 7 skipping (a.u., arbitrary units). **c, d**, Neurophysiologic findings at Nasalis muscle in repetitive nerve stimulation (RNS) recordings. **c**, Decrementing response (19.7%) induced by low-frequency (3 Hz) repetitive nerve stimulation. **d**, Incrementing response (17.8%) induced by high-frequency (20 Hz) repetitive nerve stimulation. The findings are compatible with a compromised presynaptic neuromuscular junction function. **e**, A three-generation pedigree illustrating two affected cousins, patients 4a and 4b, harboring identical pathogenic homozygous variants p.(R202H) in *UNC13A*.

**a**

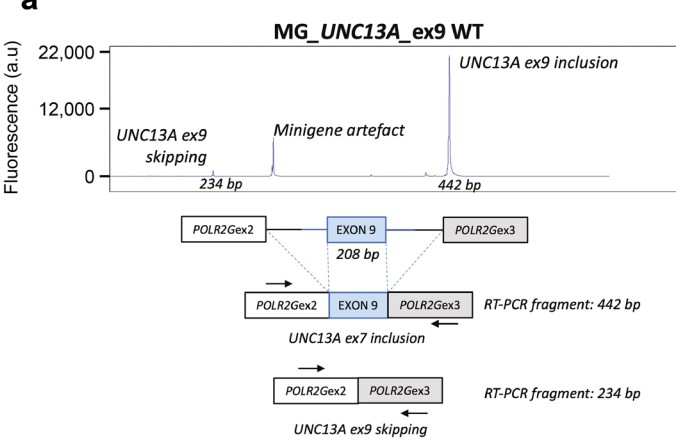
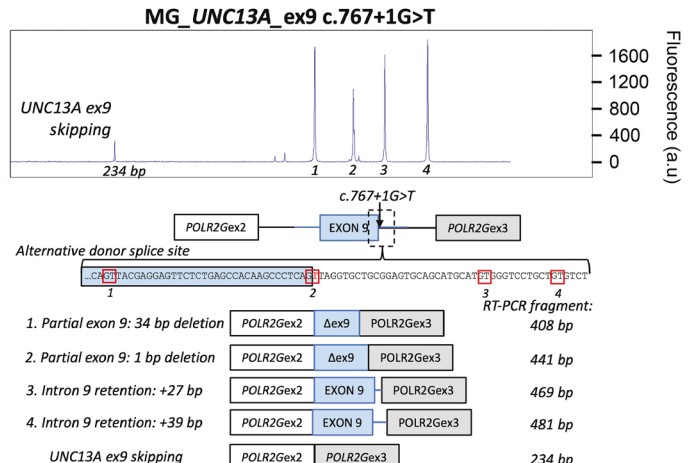

**b**

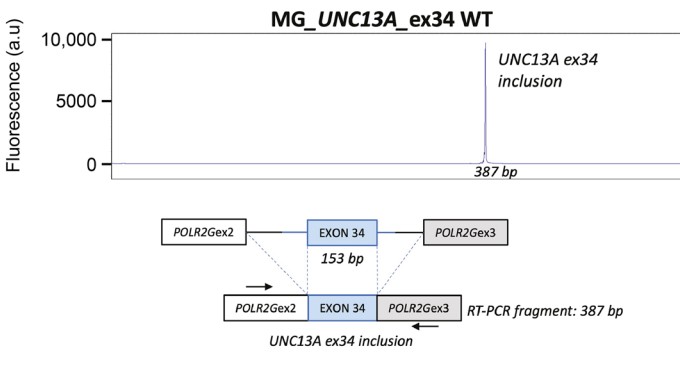
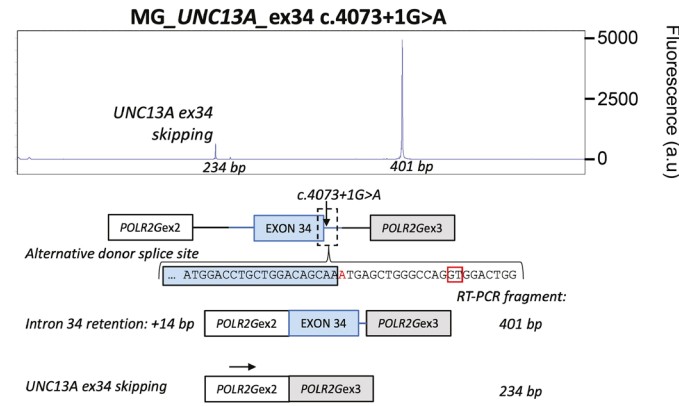

**c**

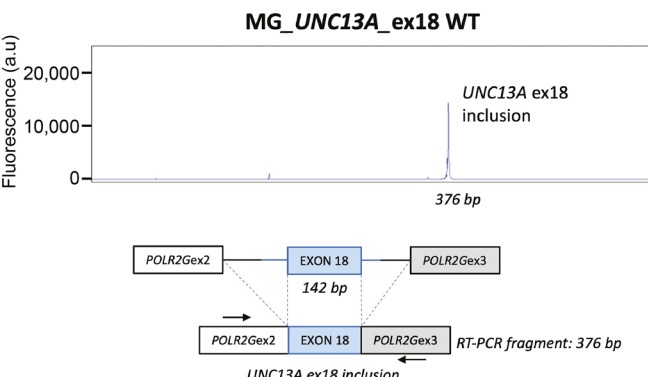
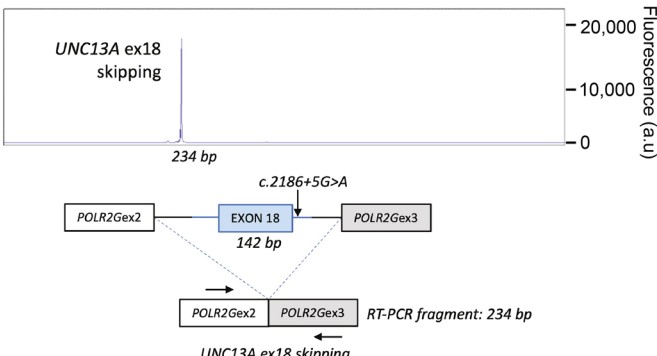

**Extended Data Fig. 3 | Minigene in vitro analysis of three *UNC13A* intronic variants. a, b**, *UNC13A* c.767+1 G > T (intron 9) and c.4073+1 G > A (intron 34) variants inherited in compound heterozygous state exert strong, deleterious impact on *UNC13A* exon 9 (**a**) and exon 34 splicing (**b**), respectively. **c**, an *UNC13A* c.2186+5 G > A variant (intron 18) inherited in a homozygous state revealed strong, deleterious impact on *UNC13A* exon 18 splicing. All events are predicted to cause mis-splicing and downstream nonsense-mediated mRNA decay. a.u., arbitrary units.

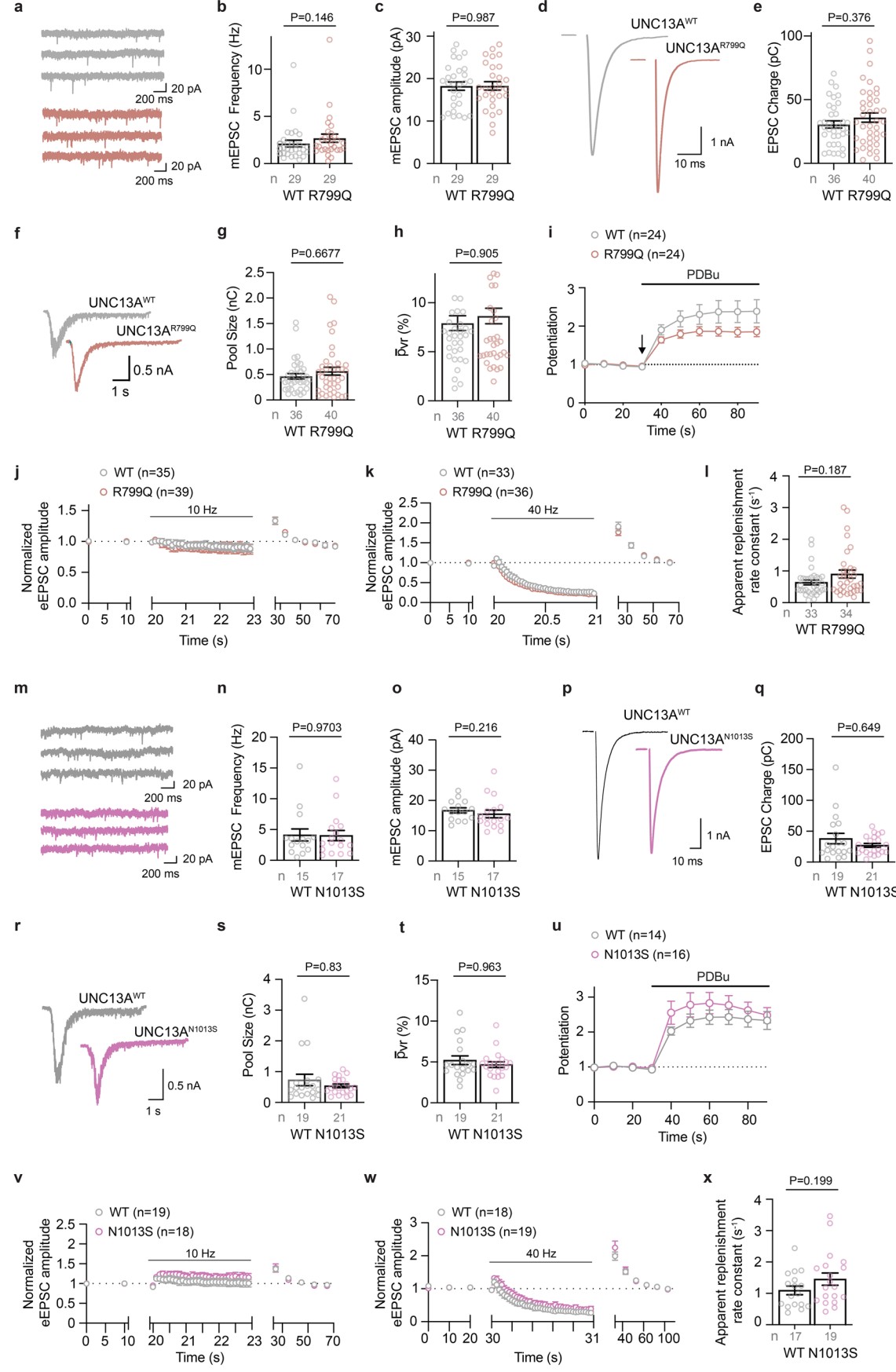

**Extended Data Fig. 4 | See next page for caption.**

**Extended Data Fig. 4 | Electrophysiological characterization of UNC13A$^{R799Q}$ and UNC13A$^{NI013S}$ in Unc13a/b DKO neurons reveals no evidence for variant pathogenicity. a-l,** Data obtained from neurons expressing UNC13A$^{WT}$ (gray) or UNC13A$^{R799Q}$ (brown). **a,** Example traces of spontaneous miniature excitatory postsynaptic currents (mEPSCs), and **b,** quantification of mEPSC frequency and **c,** mEPSC amplitude. **d,** Example traces and **e,** quantification of the charge transfer during a single action-potential evoked excitatory postsynaptic currents (eEPSCs). **f,** Example traces and **g,** quantification of the readily-releasable pool of SVs by application of hypertonic sucrose. **h,** Average vesicular release probability $\bar{p}_{vr}$. **i,** Changes in eEPSC amplitude in response to PDBu application during a 0.1 Hz action potential stimulation recorded in UNC13A$^{WT}$ (gray) or UNC13A$^{R799Q}$ (brown) expressing neurons. **j,k,** Averaged time courses of normalized eEPSC amplitudes, before, during and after a high-frequency action potential train at 10 Hz (**j**) or 40 Hz (**k**), for the indicated genotypes. **l,** Replenishment rate constant calculated based on a single-pool model[58] for neurons of the indicated genotypes. **m-x,** similar data for neurons expressing UNC13A$^{WT}$ (gray) or UNC13A$^{NI013S}$ (purple). Bars and circles in **a-h, l, m-t** and **x** represent means and values for individual neurons, respectively. Error bars, s.e.m. Statistical analysis: two-tailed Mann-Whitney test. See Supplemental table 2 and 3 for further details.

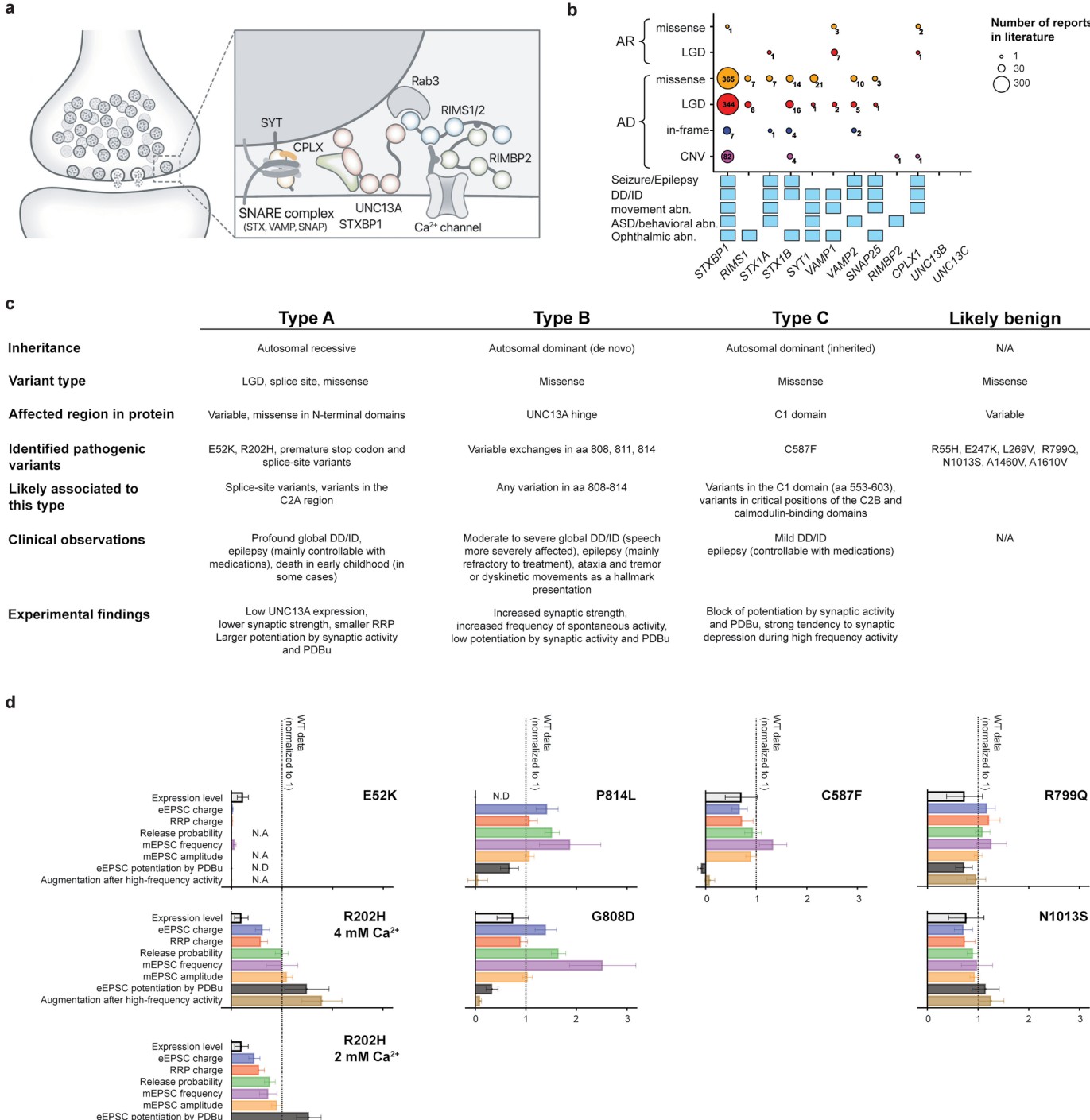

**Extended Data Fig. 5 | Landscape of the UNC13A disorder. a**, Illustrations of the synapse (left) and the core molecular machinery of the active zone (right). Variations in the genes encoding for these proteins are cause for SNAREopathy disorders. **b**, Spectrum of pathogenic variants, patterns of inheritance and associated phenotypes reported for genes encoding neuronal SNAREs and associated AZ proteins. Frequency of diverse disease-causing variants and their patterns of inheritance are shown based on the data in the Human Gene Mutation Database (HGMDv.4.1). Size of circles corresponds to the frequencies of reports in the literature with respective numbers labelled. Spectrum of associated phenotypes derived from clinical synopses in OMIM, GeneReviews and HGMD

is shown with light blue boxes for each gene. AD, autosomal dominant; AR, autosomal recessive; ASD, autism spectrum disorder; CNV, copy number variant; DD, developmental delay; ID, intellectual disability; LGD, likely gene disrupting. **c**, Summary of the key data defining the UNC13A condition, stratified according to the three subtypes defined in the present study. **d**, Summary of the key experimental findings for each of the characterized disease variants. The data is normalized to the respective control (dotted vertical lines) to allow a comparison of magnitudes and directions of variant-associated changes. SEMs were calculated by Gaussian error propagation. Data for the P814L variant are obtained from[24]. N.A; not analyzed, N.D; not determined.

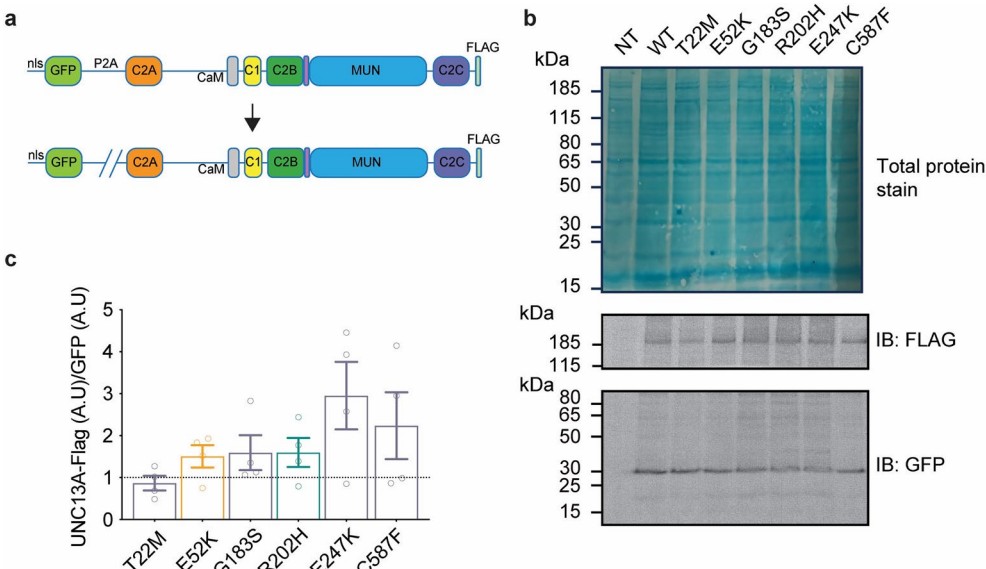

**Extended Data Fig. 6 | UNC13A disease variants do not affect UNC13A stability in HEK293FT cells. a**, A GFP-P2A-UNC13A-FLAG (top) is cleaved in cells to produce two fragments (bottom), GFP and UNC13A-FLAG. **b**, An example blot including total protein stain (up), an anti-FLAG immunoblot (IB; middle), and an anti-GFP immunoblot (down) from one of four experiments used to determine the expression level of UNC13A-FLAG with or without disease variants in comparison to that of GFP. **c**, Changes in the ratio of UNC13A-FLAG to GFP intensity were statistically not significant, which agrees with the notion that disease variants do not interfere with UNC13A stability. Circles represent four independent biological replications. Error bars represent ± SEM. Statistical anaylsis: Kruskal-Wallis test.

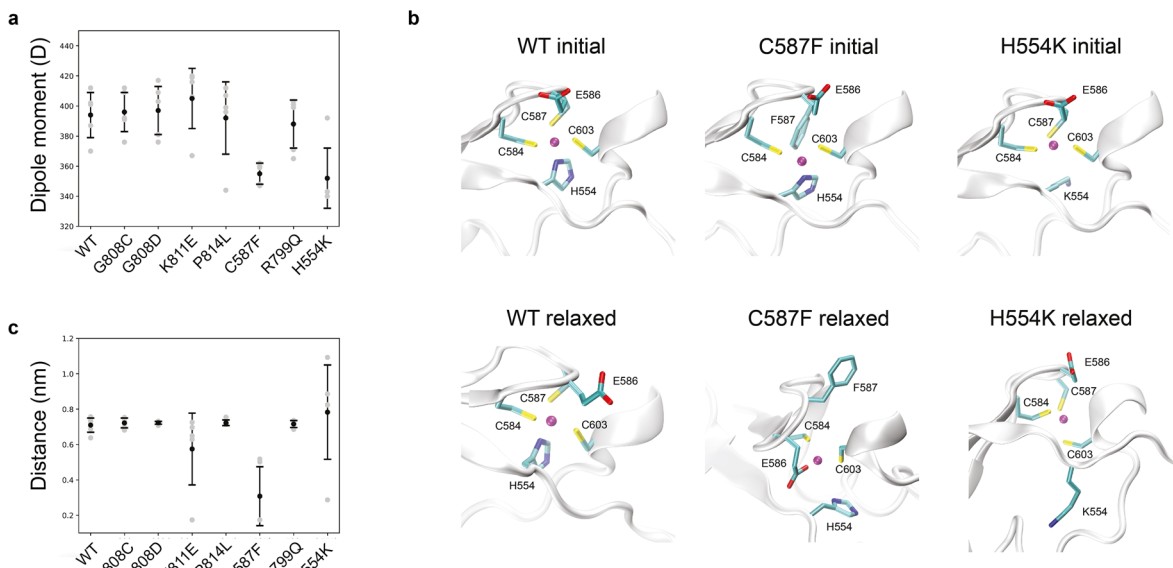

**Extended Data Fig. 7 | Molecular Dynamics (MD) simulation predicts variant-specific reorganization of the UNC13A C1 domain.** The C1 domain structure is maintained by a zinc-cluster[104]. Two $Zn^{2+}$ ions are coordinated each by one histidine and three cysteine residues[105]. C587, in which a disease variation was found, and H554 (H567 in the rodent UNC13A homolog; Fig. 5d) are $Zn^{2+}$-coordinating residues. The finding that the C587F variant has a minor effect on basal synaptic transmission properties is surprising, because knock-in mouse neurons expressing a H567K mutation that abolishes PDBu/DAG binding[14,15,49] have impaired synaptic transmission properties. To understand the cause of this difference, we compared the effect of H554K and C587F on the structural integrity of the human UNC13A C1 domain using molecular dynamics (MD) simulations. The human UNC13A models comprising the C1, C2B, and MUN domains (residues 505 to 985) were predicted with Alphafold2[89] using the UniProt sequence Q9UPW8 as input. We ran five independent 1 μs-long MD simulations for the WT protein, H554K, C587F, and for other disease variants as controls (see materials and methods). **a**, Averaged dipole moments of the C1 domain during the last 250 ns of five simulations (black dots). We did not observe unbinding of the $Zn^{2+}$ ion or unfolding of the domain within the simulation time period. **b**, Representative snapshots of the $Zn^{2+}$ ion binding pocket in the WT UNC13A C1 domain and in the UNC13A C1 domain carrying the C587F and H554K substitutions, after 1 μs MD simulations ('relaxed') compared to the initial state. The $Zn^{2+}$ ion is represented by as a blue sphere. The adjacent E586 replaces F587 in the coordination of the $Zn^{2+}$ ion, but not K554. **c**, The averaged minimal distances between $Zn^{2+}$ and E586 of the last 250 ns of five independent MD simulations (black dots). The structural and electrostatic changes observed in MD simulations support the notion that C587F exerts a different effect on the C1 domain structure than H554K, although both abolish PDBu sensitivity. In **a** and **c** the standard deviation is indicated by black bars. Averages are highlighted as gray dots.

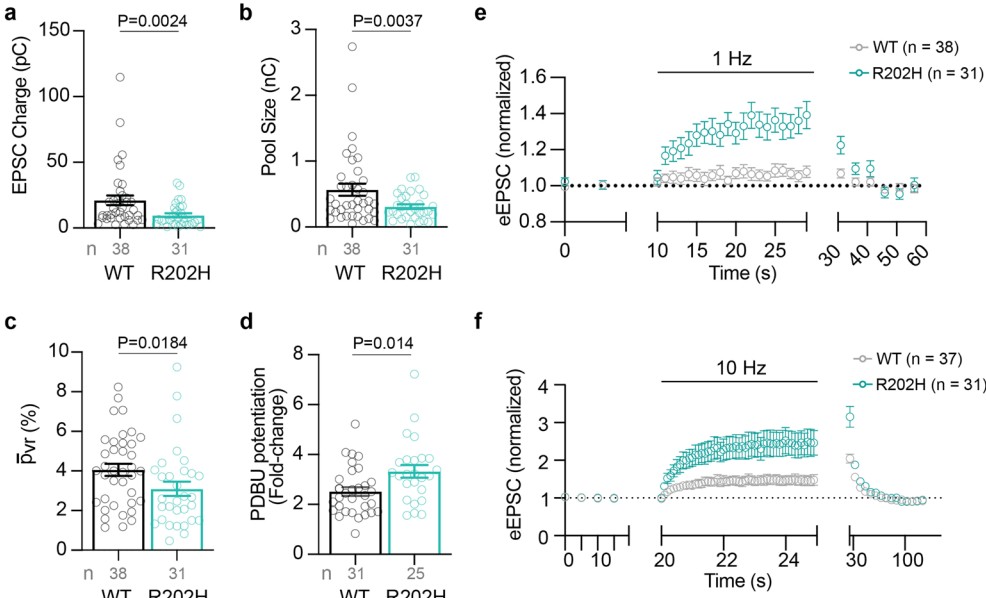

**Extended Data Fig. 8 | The UNC13A R202H disease variant leads to reduced synaptic strength and enhances short-term facilitation and augmentation.** Recordings in autaptic hippocampal neurons were made in the presence of 2 mM $Ca^{2+}$/2 mM $Mg^{2+}$ in the extracellular solution. **a**, EPSC charge, **b**, RRP size and **c**, the average vesicular release probability $\bar{p}_{vr}$ are reduced in UNC13A$^{R202H}$-expressing neurons. Note that $\bar{p}_{vr}$ was unchanged under the experimental conditions in Fig. 3. **d**, Potentiation of the EPSC amplitude by the DAG analog PDBu is stronger in UNC13A$^{R202H}$-expressing neurons. **e,f**, Stronger facilitation during and larger augmentation after AP trains at 1 Hz or 10 Hz are observed in neurons expressing UNC13A$^{R202H}$. Note that before and after the high frequency trains, EPSCs were probed at frequency of 0.2 Hz, and not at 0.1 Hz as in the rest of the manuscript. Bars and circles in **a-d** represent means and values for individual neurons, respectively. Error bars represent ± SEM. Statistical anaylsis: two-tailed Mann-Whitney test.

Reza Asadollahi, University of Greenwich, UK

# Reporting Summary

## Statistics

For all statistical analyses, confirm that the following items are present in the figure legend, table legend, main text, or Methods section.

| n/a | Confirmed | |
|---|---|---|
| ☐ | ☒ | The exact sample size (*n*) for each experimental group/condition, given as a discrete number and unit of measurement |
| ☒ | ☐ | A statement on whether measurements were taken from distinct samples or whether the same sample was measured repeatedly |
| ☐ | ☒ | The statistical test(s) used AND whether they are one- or two-sided *Only common tests should be described solely by name; describe more complex techniques in the Methods section.* |
| ☒ | ☐ | A description of all covariates tested |
| ☐ | ☒ | A description of any assumptions or corrections, such as tests of normality and adjustment for multiple comparisons |
| ☐ | ☒ | A full description of the statistical parameters including central tendency (e.g. means) or other basic estimates (e.g. regression coefficient) AND variation (e.g. standard deviation) or associated estimates of uncertainty (e.g. confidence intervals) |
| ☐ | ☒ | For null hypothesis testing, the test statistic (e.g. *F*, *t*, *r*) with confidence intervals, effect sizes, degrees of freedom and *P* value noted *Give P values as exact values whenever suitable.* |
| ☒ | ☐ | For Bayesian analysis, information on the choice of priors and Markov chain Monte Carlo settings |
| ☒ | ☐ | For hierarchical and complex designs, identification of the appropriate level for tests and full reporting of outcomes |
| ☒ | ☐ | Estimates of effect sizes (e.g. Cohen's *d*, Pearson's *r*), indicating how they were calculated |

*Our web collection on statistics for biologists contains articles on many of the points above.*

## Software and code

Policy information about availability of computer code

| | |
|---|---|
| Data collection | pCLAMP 10 software (Molecular Devices) |
| Data analysis | Electrophysiological recordings were analyzed using IgorPro (v6) and AxoGraph (v X) and organized using Graph Pad Prism 9.0.0. Imaging data was analyzed by Imaris (9.10.0),a nd organized with the KNIME Analytics Platform 4.7.1. Staistical comparisons were performed by Graph Pad Prism 9.0.0 or R 4.4.0. For illustration,representative images were processed with the FIJI software package version 2.14.0/1.54f. MD simulations: the following software packages were used: VMD 1.93,G romacs 2021.2,AlphaFold2 (01 JUL 21),M atlibplot 3.1.3, MDAnalysis 2.4.2. |

For manuscripts utilizing custom algorithms or software that are central to the research but not yet described in published literature, software must be made available to editors and reviewers. We strongly encourage code deposition in a community repository (e.g. GitHub). See the Nature Portfolio guidelines for submitting code & software for further information.

## Data

Policy information about availability of data

All manuscripts must include a data availability statement. This statement should provide the following information, where applicable:

- Accession codes, unique identifiers, or web links for publicly available datasets
- A description of any restrictions on data availability
- For clinical datasets or third party data, please ensure that the statement adheres to our policy

Data availability statement: All data supporting the findings of this study are available within the paper and in Source Data Tables 1-4. MD simulation data are available in https://zenodo.org/uploads/14554776. All other primary data will be made available on request from the corresponding authors. Publicaly available databases used in this study include GeneMatcher (https://genematcher.org), MetaDome web server (https://stuart.radboundumc.nl/metadome/), ClinVAR (https://www.ncbi.nlm.nih.gov/clinvar), Genome Aggregation Database (gnomAD; http://gnomad.broadinstitute.org), OMIM (http://www.omim.org), UCSC Genome Browser (https://genome.ucsc.edu), Ensembl (https://www.ensembl.org), and UniProt (https://www.uniprot.org).

Code availability statement: Code has not been developed in this manuscript.

## Research involving human participants, their data, or biological material

Policy information about studies with human participants or human data. See also policy information about sex, gender (identity/presentation), and sexual orientation and race, ethnicity and racism.

| | |
|---|---|
| Reporting on sex and gender | The sex of identified patients is indicated in supplemental table 1, no further sex or gender-related analysis was made. |
| Reporting on race, ethnicity, or other socially relevant groupings | We do not report on the race, ethnic affiliation, or other socially relevant groupings in our patient cohort. |
| Population characteristics | Our study population includes patients with neurodevelopmental features, ranging in age from 11 months to 32 years, who underwent exome or genome sequencing on a clinical or research basis at academic institutes or diagnostic labs worldwide. Informed consent was obtained from parents or guardians of the affected individuals, with approval from local institutional review boards. De-identified genetic variants, clinical data, and facial photographs (shown in Figures 3-5 and Supplementary Figures 1 and 2) were obtained from collaborating institutions based on these informed consents. |
| Recruitment | Individuals included in this study underwent exome or genome sequencing on diverse sequencing platforms on a clinical or research basis in academic institutes or diagnostic labs worldwide. Using GeneMatcher1 and personal communication with colleagues enabled us to assemble clinical and genetic details on a total of 48 individuals with de novo or inherited heterozygous or biallelic variants in UNC13A (NM_001080421.3), including two previously published cases harboring c.154G>A, p.(E52K) [case 74 of Lionel et al., 20182], and c.4379C>T, p.(A1460V) [case UPN-0740 of 3] variants with further follow up information. |
| Ethics oversight | This study was performed as part of a research study approved by the ethics commission of the Canton of Zurich (ID PB_2016-02520 [SIV 11/09]). In addition, for each patient, ethical approvals and informed consent forms from parents or guardians were obtained by the respective research teams and institutions for data use and publication, including photographs or videos where applicable. Participants did not receive compensation. |

Note that full information on the approval of the study protocol must also be provided in the manuscript.

## Field-specific reporting

Please select the one below that is the best fit for your research. If you are not sure, read the appropriate sections before making your selection.

☒ Life sciences          ☐ Behavioural & social sciences          ☐ Ecological, evolutionary & environmental sciences

For a reference copy of the document with all sections, see nature.com/documents/nr-reporting-summary-flat.pdf

## Life sciences study design

All studies must disclose on these points even when the disclosure is negative.

| | |
|---|---|
| Sample size | Sample size calculations for each of the recorded parameters were not performed prior to these experiments. In general, for the electrophysiological experiments presented, we were interested in detecting large and therefore likely biologically significant effects, i.e. differences in sample means of 0.8 to 1 times SD, corresponding to a 'Large effect' according to Cohen. Sample sizes (n) to resolve such effects are >= 17-26 for each group for a statistical power of 0.8 and a significance level of 0.05. For all experiments, n was >=19." |
| Data exclusions | In electrophysiological experiments, all data which fulfilled the quality criteria (e.g. leak current etc) was included in the analysis. No 'outliers detection' or related procedures were applied. |
| Replication | Data was collected from two or more cultures. In the vast majority of experiments, data was independently obtained by two experimenters |

| Replication | (Figure 3 - in different laboratories, in Figures 4-7 by different experimenters), to ensure reproducibility of the observations. Approximately equal numbers of WT and Munc13-1-variant recordings were obtained during each measurement day. |
| Randomization | Randomization was not used in this study, as it is not relevant in the experiments described here. |
| Blinding | Blinding was not used in this study. Instead, for each of the variants, recordings were performed by two or three independent experimenters and across two or three different labs, and all phenotypes were confirmed. |

# Reporting for specific materials, systems and methods

We require information from authors about some types of materials, experimental systems and methods used in many studies. Here, indicate whether each material, system or method listed is relevant to your study. If you are not sure if a list item applies to your research, read the appropriate section before selecting a response.

### Materials & experimental systems

| n/a | Involved in the study |
|---|---|
| ☐ | ☒ Antibodies |
| ☐ | ☒ Eukaryotic cell lines |
| ☒ | ☐ Palaeontology and archaeology |
| ☐ | ☒ Animals and other organisms |
| ☒ | ☐ Clinical data |
| ☒ | ☐ Dual use research of concern |
| ☒ | ☐ Plants |

### Methods

| n/a | Involved in the study |
|---|---|
| ☒ | ☐ ChIP-seq |
| ☒ | ☐ Flow cytometry |
| ☒ | ☐ MRI-based neuroimaging |

## Antibodies

| Antibodies used | Antibody; Source; Dilution used; RRID; Identifier; clone ID (when relevant)<br>Ab 1. Mouse monoclonal GFP; Merck Millipore; 1:250; AB_94936; MAB3580, Clone ID: N/A<br>Ab 2. Rabbit polyclonal VGLUTl; Synaptic Systems; 1:1000; AB_887877; 135 302<br>Ab 3. Guinea pig polyclonal Shank 2; Synaptic Systems; 1:250; AB_2619861; 162 204<br>Ab 4. Chicken polyclonal MAP2; Novus Biologicals; 1:1000; AB_2138178; NB300-213<br>Secondary Abs for immunostaining<br>Ab 5. Goat anti-Mouse Alexa 488; Thermo Fisher; 1:2000; AB_2534088; A11029<br>Ab 6. Goat anti-Rabbit Alexa 633; Thermo Fisher; 1:2000; AB_141419; A21071<br>Ab 7. Goat anti-Guinea Pig Alexa 568; Abeam; 1:2000; AB_2864763; Abl 75714<br>Ab 8. Goat anti-Chicken Alexa 405; Abeam 1:1000; AB_2890171; Ab175674<br>Primary antibodies for Western Blot analysis<br>Ab 9. Polyclonal rabbit anti-FLAG; Sigma-Aldrich; 1:2000; AB_ 439687; F7425<br>Ab 10. Monoclonal mouse anti- Green Fluorescent Protein (1E4); Enzo Life Sciences; 1:1000; A11. ADI-SAB-500-E; Clone ID: 1E4<br><br>Secondary antibodies for Western Blot analysis<br>Ab 12. Peroxidase AffiniPure Goat Anti-Mouse lgG (H+L), Jackson immunoResearch; 1:5000; AB_2307392; 115-035-146<br>Ab 13. Peroxidase AffiniPure Goat Anti-Rabbit lgG (H+L), Jackson immunoResearch; 1:30000 AB_2307391; 111-035-144 |
| Validation | Ab 1. Mouse monoclonal GFP; Merck Millipore; 1:250; AB_94936; MAB3580<br>Recognizes a protein tag<br>Website information: Antibody validated for use in ELISA, IC, IH & WB.<br>In this study: immunostaining signal was absent in a non-transfected control (Figure 2c, 2e)<br><br>Ab 2. Rabbit polyclonal VGLUT1; Synaptic Systems; 1:1000; AB_887877; 135 302<br>Website information: Antibody was validated using KO samples, citations can be found in https://sysy.com/product/135302#list<br><br>Ab 3. Guinea pig polyclonal Shank 2; Synaptic Systems; 1:250; AB_2619861; 162 204<br>Website information: Antibody was validated using KO samples (PMID: 2997098)<br><br>Ab 4. Chicken polyclonal MAP2; Novus Biologicals; 1:1000; AB_2138178; NB300-213<br>Website information: Knockdown Validated (PMID: 32294442). More publications in https://www.novusbio.com/products/map2-antibody_nb300-213?srsltid=AfmBOorEtC2sqRWpEaMtjX36c2y8L3tkHsKuvStTlAsTiKx6DkunwKBB#reviews-publications<br><br>Ab 9. Polyclonal rabbit anti-FLAG; Sigma-Aldrich; 1:2000; AB_ 439687; F7425<br>Recognizes a protein tag<br>In this study: validated in this study by targeting a non-transfected control using western blot (Supplementary figure 4b)<br><br>Ab 10. Monoclonal mouse anti- Green Fluorescent Protein (1E4); Enzo Life Sciences; 1:1000; ADI-SAB-500-E<br>Recognizes a protein tag<br>In this study: validated in this study by targeting a non-transfected control using western blot (Supplementary figure 4b) |

## Eukaryotic cell lines

Policy information about cell lines and Sex and Gender in Research

| | |
|---|---|
| Cell line source(s) | HEK293T (ATCC-CRL-3216), human embryonic kidney, commercially available |
| Authentication | The cell line was not authenticated. |
| Mycoplasma contamination | Mycoplasma contamination was not detected. |
| Commonly misidentified lines (See ICLAC register) | Not relevant |

## Animals and other research organisms

Policy information about studies involving animals; ARRIVE guidelines recommended for reporting animal research, and Sex and Gender in Research

| | |
|---|---|
| Laboratory animals | Mice: We used mice at embryonic day 18 (E18) to prepare primary hippocampal neuronal cultures<br>Strain: Unc13a/b double knock-out mice (MGI Unc13atm1Bros and Unc13btm2Bros) - made in the lab of co-authors of this study (Dr. Nils Brose). Adult mice were kept under IVC/SPF conditions, at 12h/12h light/dark cycle, at room temperature of 22 +/-2°C, and humidity levels of 55 +/-10%.<br>C. elegans: Control strains used: N2, unc-13(nu641). unc-13(nu641) harbors a C-terminal mScarlet in the unc-13 locus. CRISPRmodified strains were outcrossed at least four times and the relevant genomic region was sequenced to confirm the target mutation. |
| Wild animals | No wild animals were used |
| Reporting on sex | Mice: Sex was not considered and both female and male embryonic mice were used to make cultures<br>C. elegans: hermaphrodite worms were used |
| Field-collected samples | The study did not invlove samples collected from the field. |
| Ethics oversight | The use of the Unc13a/b knockout mice were approved by the responsible local government organizations in Germany (Niedersächsisches Landesamt für Verbraucherschutz und Lebensmittelsicherheit; 33.19-42502-04-15/1817 and 33.19-42502-04-20/3589, and Landesamt für Gesundheit und Soziales; G106/20) |

Note that full information on the approval of the study protocol must also be provided in the manuscript.

## Plants

| | |
|---|---|
| Seed stocks | Report on the source of all seed stocks or other plant material used. If applicable, state the seed stock centre and catalogue number. If plant specimens were collected from the field, describe the collection location, date and sampling procedures. |
| Novel plant genotypes | Describe the methods by which all novel plant genotypes were produced. This includes those generated by transgenic approaches, gene editing, chemical/radiation-based mutagenesis and hybridization. For transgenic lines, describe the transformation method, the number of independent lines analyzed and the generation upon which experiments were performed. For gene-edited lines, describe the editor used, the endogenous sequence targeted for editing, the targeting guide RNA sequence (if applicable) and how the editor was applied. |
| Authentication | Describe any authentication procedures for each seed stock used or novel genotype generated. Describe any experiments used to assess the effect of a mutation and, where applicable, how potential secondary effects (e.g. second site T-DNA insertions, mosiacism, off-target gene editing) were examined. |

