## [Peer Review File · Nature Genetics]

Pathogenic UNC13A variants cause a neurodevelopmental syndrome by impairing synaptic function

Corresponding Author: Dr Noa Lipstein

Version 0:

Decision Letter:

25th Jul 2024

Dear Noa,

Your Article, "A Novel Neurodevelopmental Syndrome Caused by Pathogenic UNC13A Variants that Impair Synaptic Function" has now been seen by 3 referees. You will see from their comments below that while they find your work of interest, some important points are raised. We are interested in the possibility of publishing your study in Nature Genetics, but would like to consider your response to these concerns in the form of a revised manuscript before we make a final decision on publication.

In brief, the three referees are all very supportive of this work and strike notably positive tones throughout. There are a few requests to improve a revision, but in our reading they are not asking for any major further expansion of the work and the requests seem by and large easily addressable. We therefore hope you and your co-authors will be motivated to respond to them in full.

To guide the scope of the revisions, the editors discuss the referee reports in detail within the team, including with the chief editor, with a view to identifying key priorities that should be addressed in revision and sometimes overruling referee requests that are deemed beyond the scope of the current study. We hope that you will find the prioritized set of referee points to be useful when revising your study. Please do not hesitate to get in touch if you would like to discuss these issues further.

We therefore invite you to revise your manuscript taking into account all reviewer and editor comments. Please highlight all changes in the manuscript text file. At this stage we will need you to upload a copy of the manuscript in MS Word .docx or similar editable format.

*2) If you have not done so already please begin to revise your manuscript so that it conforms to our Article format instructions, available

http://www.nature.com/ng/authors/article_types/index.html here

*3) Include a revised version of any required Reporting Summary: <https://www.nature.com/documents/nr-reporting-summary.pdf>

Please be aware of our <https://www.nature.com/nature-research/editorial-policies/image-integrity> guidelines on

digital image standards.

Link Redacted

Nature Genetics is committed to improving transparency in authorship. As part of our efforts in this direction, we are now requesting that all authors identified as 'corresponding author' on published papers create and link their Open Researcher and Contributor Identifier (ORCID) with their account on the Manuscript Tracking System (MTS), prior to acceptance. ORCID helps the scientific community achieve unambiguous attribution of all scholarly contributions. You can create and link your ORCID from the home page of the MTS by clicking on 'Modify my Springer Nature account'. For more information please visit please visit www.springernature.com/orcid.

Sincerely,

Michael Fletcher, PhD
Senior Editor, Nature Genetics
ORCID: 0000-0003-1589-7087

Referee expertise:

Referee #1: functional genomics; neurogenetics

Referee #2: neuronal biology, including synapses

Referee #3: ALS/FTD, clinical neurology

Reviewers' Comments:

Reviewer #1:

Remarks to the Author:

The study by Asadollahi et al describes a series of genetic variants in UNC13A in 48 individuals that lead to neurodevelopmental disorders. The authors systematically characterize the consequences of these variants in an allelic series of 7 representative mutations on synapse formation and function in cultured mouse hippocampal neurons and at the nematode neuromuscular junction. The authors demonstrate that UNC13A variants affect synaptic expression levels and synaptic transmission in a variant-specific manner, with different underlying molecular mechanisms: reduction of UNC13A expression levels, gain of function in synaptic transmission, altered synaptic plasticity and protein regulation. Based on these functional analyses, the authors propose three pathogenic mechanisms and classify patients accordingly.

This study uncovers a fascinating complexity in disease mechanisms for genetic variants in this gene, with classical heterozygous loss of function mutations, both de novo and inherited, but also biallelic recessive mutations, that cause haploinsufficiency, but also gain of function phenotypes and neomorphs. The functional analyses are state of the art, using primary neurons and in vivo models (nematodes). Consequently, the precision of the functional phenotyping is exceptional with well-defined and selective effects on expression levels, synaptic transmission, -plasticity and probably transport to the synapse. The proposed discrimination of variants into three classes of mutations is very convincing. This new classification is important for future patient stratification. The manuscript is very well written. The study also delivers new information on the physiological functions of this protein (e.g. the critical role of the 'hinge' domain, with interesting new gain of function phenotype). Taken together, this study contains all the ingredients for a hallmark paper: a detailed genetic characterization, excellent functional analysis and a classification of three subtypes that will guide stratification of future cases and therapy design.

On the other hand, the translation of findings in mouse neurons to the human situation is underdeveloped, the genotype-phenotype relations can be defined more precisely, the organization of the manuscript can be improved and the Discussion is a bit limited in scope.

(1) Translation: The manuscript contains little guidance how to extrapolate findings made in prototypical mouse synapses towards the human phenotypes. Previous studies in mouse neurons show that UNC13B is redundant with UNC13A function in some neurons, but not others, giving rise to selective deficits in excitatory neurons. Hence, phenotypes observed in neurons where both UNC13A and B have been inactivated, are expected to translate in a different manner in human

neurons that do or do not express UNC13B and (often) one normal UNC13A allele. The aspect of selective deficits in excitatory neurons is probably important to better understand human symptoms (excitation/inhibition balance) and should be discussed. The known expression pattern of both genes in the human brain can help to further predict disease mechanisms at the systems level. Finally, it would be good to emphasize it is not known how phenotypes observed in double null mutant mouse neurons impact human neurons that express two healthy UNC13B and (in most cases) one healthy UNC13A allele.

(2) Genotype-phenotype relations: The first part of the Discussion (line 509-534) contains valuable leads on how different classes of mutations give rise to distinct symptoms. However, there is no clear synthesis on what are the emerging differences in the symptom spectrum among the three classes. This is a key issue that will interest the majority of the readership and should be defined in the Discussion and probably also the Abstract and end of the Introduction.

(3) Data presentation can be improved. It is hard to compare different mutants without memorizing the amino acids. Why are similar mutants kept apart in Fig 2? Even if they were tested in separate experiments and have their own control groups, they can still be together. This makes it easier to conclude how similar they behave. At the very least make the scales the same. Organization of Fig 5-6 is not great. Now Fig 5 combines a family tree of one mutation with synaptic plasticity data from three and only in Fig 6 we find the basic transmission data of the mutation of the family tree in Fig 5. Why not present these two together in Fig 5 and move synaptic plasticity data from three mutations to Fig 6? It would be great to add some summary data at the end of the manuscript, bringing (normalized?) key data of all mutations studied together so the readership can evaluate the defining differences in synaptic parameters in a single figure/panel.

(4) The discussion is currently oriented towards molecular mechanisms and delivers relatively little scope (except the comparison between effects of UNC13A mutations early and later in life, which is great but also long). It would be great to touch upon a few more general topics, e.g. (i) genotype-phenotype relations (see above), (ii) is the classification into three groups definitive or are more classes predicted to emerge based on what we know about UNC13A function and the fact that currently known mutations are both in tolerant and intolerant regions? (iii) a comparison to developmental disorders caused by mutations in genes that operate in the same molecular machine as UNC13A and their fascinating genetic complexity.

Reviewer #2:

Remarks to the Author:

The (many) authors of an international collaboration went through a concerted effort which resulted in describing a novel neurodevelopmental syndrome(s) linked to germline coding or splice site variants in the UNC13A gene, which can follow autosomal dominant (de novo or inherited heterozygous pathogenic variants) or autosomal recessive (biallelic pathogenic variants) inheritance patterns. The syndrome presents with a spectrum of symptoms, including developmental delay or intellectual disability, epilepsy, tremor, dyskinetic movements, and, in severe cases, early childhood mortality.

They use a broad array of assays to evaluate UNC13A protein stability and abundance at synapses, the strength and plasticity of neurotransmitter release using electrophysiological recordings in mouse hippocampal excitatory neurons, movement in *C. elegans* knock-in worms, synaptic responses to second messenger signaling, and structural changes through molecular dynamic modeling. Thus, they describe three distinct pathogenic mechanisms:

- Reduction in synaptic strength due to decreased UNC13A protein levels
- Increased neurotransmission resulting from a gain of function in UNC13A
- Impaired synaptic response to second messenger signaling

Based on this analysis (genotype, phenotype, physiological phenotype), they further classify UNC13A syndrome subtypes.

I am in the rare situation of having little to nothing to criticize here. The identified mutations significantly advance our understanding of UNC13A biology in humans and might potentially lay a foundation for developing therapeutic interventions. Clearly, this material apart from its clinical implications will also be of importance concerning our mechanistic understanding of Unc13 function in a generic manner. This also given that a bulk of the physiological information is retrieved from autaptic recordings, and much remains to be learned concerning the truly in vivo impact of these mutations (e.g. in the transmission across frequency space). Equally, it will (in the future) be important to address at which synapses these mutations ultimately execute their pathologically relevant dysfunctions. Still, given the major progress this paper marks, I do unreservedly suggest publication of the manuscript in its current form.

Reviewer #3:

Remarks to the Author:

This is very interesting study showing that pathogenic variants in UNC13A are (1) either biallelic or monoallelic with de novo or familial inheritance and (2) that pathogenic variants affect different UNC13A protein domains that participate in defining different properties of synaptic transmission, with evidence for at least three independent mechanisms: loss of function, gain-of-function (increased synaptic transmission) and dysregulation of transmission.

In OMIM two early onset syndromes are linked to UNC13A variants, albeit not uncontroversial: a congenital myasthenic syndrome and a dyskinetic movement disorder associated with delayed development and behavioral abnormalities. Could the authors discuss how their reported syndromes relate to these prior reported syndromes, and if not, how we should interpret these prior reports?

The authors frequently refer to the recent ALS/FTD literature on UNC13A cryptic exon splicing induced by TDP-43 mislocalisation. Indeed, the question is, if this study helps to understand the mechanisms at play in ALS/FTD, since this has not yet been resolved. The cryptic exon is located between exon 20 and 21, so sort of right in the "UNC13A Hinge" as the authors call it. Still, we "think" the effect in ALS/FTD is "loss of function" as UNC13A is misspliced and the aberrant mRNA species are degraded by non-sense mediated decay. Nevertheless, we know that in ALS/FTD glutamate excitotoxicity is one of many pathogenic mechanisms, so I wondered if the authors could reflect on which of their presented mechanisms could be at play in ALS/FTD? It is very intriguing to see that coding mutations in UNC13A could lead to a toxic gain of increased synaptic transmission, but how can one this reconcile with the presumed loss of function following missplicing of UNC13A? It would be great if the authors could reflect on this, and dedicate a paragraph to this.

I have one methodological question: how exactly did the authors quantify synapse numbers and in which (non)human systems? This is a bit unclear from the methods section. Is it done using antibodies against SHANK2 and VGLUT1? In *C. elegans*?

Textual:

The introduction has a strange start with "In a ground-breaking screen for genes controlling *C. elegans* behaviour¹, uncoordinated (UNC) strain number 13 (*unc-13*) was scored as severely affected." I would start with a combination of the second sentence, e.g. In a groundbreaking.... worms with a"

Version 1:

Decision Letter:

Our ref: NG-A65794R

6th Dec 2024

Dear Noa,

Thank you for submitting your revised manuscript "A Novel Neurodevelopmental Syndrome Caused by Pathogenic UNC13A Variants that Impair Synaptic Function" (NG-A65794R). It has now been seen by the original referees and their comments are below. The reviewers find that the paper has improved in revision, and therefore we'll be happy in principle to publish it in Nature Genetics, pending minor revisions to satisfy the referees' final requests and to comply with our editorial and formatting guidelines.

Sincerely,

Michael Fletcher, PhD
Senior Editor, Nature Genetics
ORCID: 0000-0003-1589-7087

Reviewer #1 (Remarks to the Author):

the authors provide adequate responses to all my prior concerns and made adequate adjustment to the manuscript. The authors deserve a big complement on this excellent study

Reviewer #2 (Remarks to the Author):

I am impressed by the revision and suggest publication unreservedly.

Reviewer #3 (Remarks to the Author):

The authors have addressed all reviewers comments more than adequately, I appreciate the additions in the Discussion further placing the findings into context.

I only noticed on typo: Extended data Figure 8: the table says "Inheritance", needs to be "Inheritence". Very well done all in all.

*

*

6C *DA=JIDABK<CLBCGCK<DAN?<BDBL<C<<OCJD?*M*?HK<BCKCLB<C>*DAC*O=JH<PKBND*=PP?K>BJR*
D?DAC?J<DKHPD<NBJD<*K=B<C>*\S*DAB<*KAB<BCKD=JDB=<DDB?LDANKC<CJD=DB?J*?M*?HK*
MBJ>BJR<

*

;XF*.K=J<@=DB?JW*.AC*O=JH<PKBND*P?JD=BJ<*@BDD@C*RHB>=JPC*A?G*D?*CUDK=N?@=DC*MBJ>BJR<*<
O?H<C* <SJ=N<C<* D?G=K><* DAC* AHO=J* NACJ?DSNC<E* +KCLB?H<* <DH>BC<* BJ* O?H<C* JCHK?J<* <A?<
Y!9XZ/*B<*KC>HJ>=JD*GBDA*Y!9XZ\$*MHJPDB?J*BJ*<?OC*JCHK?J?QABB<Q*RBLBJR*KB<C*D?* <C@CPDBL<
>CMBPBD<*BJ*CUPBD=D?KS*JCHK?J<E*4CJPCQ*NACJ?DSNC<?*?\<CKLC>*BJ*JCHK?J<*GACKC\?DA*Y!9XZ\$*=<
\CCJ*BJ=PDBL=DC>Q*=KC*CUNCPDC>*D?*DK=J<@=DC*B<*=*>BMMCKCJD*O=JJK*BJ*AHO=J*JCHK?J<*D<A<
CUNKC<<*Y!9XZ/*=J\$;?MDCJF?*JC*J?KO=@*Y!9XZ\$*=@@C@CE*.AC*=<NCPD?*M*<C@CPDBLC*>CMBPBD<*<
JCHK?J<*B<*NK?\=@S*BON?KD=JD*D?*CDDCK*HJ>CK<D=J>*AHO=J*<SOND?O<*;CUPBD=DB?JcBJAB\BDB?J<
<A?H@>*\C>B<PH<AC>E?GJ*CUNKC<<B?J*N=DDCKJ?*M*\DA*RCABO*BJ*ABJ*P=J*AC@N*D?*MHKDACK*
NKC>BPD*>B<C=<C*OCPA=JB<O<*=D*DAC*<S<BGC@<@<QLB@<E?H@>*\C*R??>*D?*CONA=<BaC*BD*B<*J?D*IJ<
A?G*NACJ?DSNC<?*?\<CKLC>*BJ*>?H\@C*JH@@*OHD=JD*O?H<C*JCHK?J<*BON=PD*AHO=J*JCHK?J<*DA=J<
DG?*AC=@DAS*Y!9XZ/*=J\$;?MDCJF?*JC*J?KO=@*Y!9XZ\$*=@@C@CE

*

*

6C* MH@@S* =RKCC* GBDA* DAC* KCLBCGCK<BON?BDAB*NBJDE* 6C* A=LCD<C>CPDB?J? * DAC*
O=JH<PKBND<CKC*CC@=\?K=DC?*J*DABWD?NBP

*

-JDK?>HPD<@BJC>X#(FW

*

]B<PH<<B?@BJC<*#)(FW

*

*

/=<C>*?J*DAB<[HC<DB<C?<C<<C>*Y!9XZ/*CUNKC<<B?J*>=D*MK?O*<BJR<CJPB<@<E<CE<
\$@DA?HRCHY!9XZ/* CUNKC<<B?J*P=J* @=KRC@S*P?ONCJ<=DC*MYXZ\$RCJCPBB*PH@DHKC>*

! "\$ % & ' () * + , - . : ; < = > ? @ [\] ^ _ ` { | } ~ ! " # \$ % & ' () * + , - . : ; < = > ? @ [\] ^ _ ` { | } ~

hippocampal inhibitory neurons, according to the Allen Brain Atlas Transcriptomics explorer (Human M1 10x), UNC13B transcript expression levels appear lower in inhibitory neurons than in excitatory neurons. Based on this, we decided to avoid making additional predictions beyond those listed above. To date, data on the degree of UNC13B protein expression in neuronal subtypes of the mouse or human brain is largely missing. Expression of mRNA, while likely indicative of protein expression, may still result in partial expression of UNC13B at the synapse level, as in the case of cultured hippocampal excitatory neurons. We believe that independent studies on the expression of UNC13B will be needed to provide more accurate predictions.

(2) Genotype-phenotype relations: The first part of the Discussion (line 509-534) contains valuable leads on how different classes of mutations give rise to distinct symptoms. However, there is no clear synthesis on what are the emerging differences in the symptom spectrum among the three classes. This is a key issue that will interest the majority of the readership and should be defined in the Discussion and probably also the Abstract and end of the Introduction.

We would like to thank the reviewer for pointing this out. In the introduction and in the respective parts of the discussion section, we now define hallmark clinical features for each of the syndrome subtypes:

- Myasthenic presentation as characteristics of the Type A condition
- Ataxia, tremor, or a dyskinetic movement disorder as a hallmark of the Type B condition
- Mild symptoms and heritability in the Type C condition
- We also note differences in the degree to which epileptic seizures can be controlled by medication in all three conditions.

In addition to the respective text passages in the discussion and introduction sections (see below), we also included a new summary figure (Extended Figure 8), which provides a comparative overview of the clinical, genetic, and experimental findings in our study for the three conditions.

Introduction (Lines 93-102): Patients with the Type A condition present with profound global developmental delay and early-onset seizures. These patients harbour biallelic loss-of-function missense, truncating or splice site variants, that lead to a >50% reduction of UNC13A expression in experimental models, and to a severe reduction in neurotransmission. Patients with the Type B condition exhibit developmental delays, particularly in speech acquisition, and ataxia, tremor, or a dyskinetic movement disorder as hallmarks of the condition. These patients harbour de novo missense variants that result in a gain of UNC13A function, leading to enhanced neurotransmission. The Type C condition is caused by a familial heterozygous missense variant that results in altered regulation of UNC13A function. The patients are mildly affected, exhibiting learning difficulties to moderate ID and seizures.

Discussion (Lines 379-387): These variants severely affect developmental milestones leading to profound GDD in all cases, and in some cases cause death in early childhood due to respiratory failure after pneumonia (Fig. 1, Supplementary table 1). In addition, all patients develop early-onset seizures that mostly responds to antiepileptic treatment (Supplementary table 1). Based on a previously reported case¹⁰ harbouring a homozygous stop codon variant, who was diagnosed with fatal myasthenia according to electromyography findings, and on our patients 3 and 7 (Supplementary table 1, Extended data figure 2 and 3), we suggest considering myasthenic presentation in Type A patients. To clarify whether this is a typical presentation requires further investigations.

(3) Data presentation can be improved. It is hard to compare different mutants without memorizing the amino acids. Why are similar mutants kept apart in Fig 2? Even if they were tested in separate experiments and have their own control groups, they can still be together. This makes it easier to conclude how similar they behave. At the very least make the scales the same.

To facilitate data comparison, we now

- included a new summary figure (Extended Figure 8), where the data obtained in this study are presented as summary bar graphs, trusting that this enables a quick comparison of conditions as well as an overall evaluation of the range of changes per condition. In the new Extended

Figure 8, we normalize data for each condition to the respective control measurements, summarizing magnitude and direction of change for comparison.

- re-scaled the majority of graphs in the manuscript to have an identical scale.
- reorganized Figure 2 to enable a better comparison between conditions.
- combined figures 5 and 6 into a single figure (see below).

We trust that these changes substantially improved the data presentation in the manuscript and thank the reviewer again for this comment.

Organization of Fig 5-6 is not great. Now Fig 5 combines a family tree of one mutation with synaptic plasticity data from three and only in Fig 6 we find the basic transmission data of the mutation of the family tree in Fig 5. Why not present these two together in Fig 5 and move synaptic plasticity data from three mutations to Fig 6?

We followed the reviewer's suggestion and re-organized the figure. We present all relevant data for the C587F mutation in one figure (Figure 5). Because this variant interferes with DAG/PDBu sensitivity, we also include in this figure additional data for the DAG/PDBu sensitivity of additional variants examined in this study, and make sure the data are presented in a way that enables visual comparison. All synaptic short-term plasticity data are now included in Figure 6. We hope this will increase the readability of the manuscript and improve the flow of the data presentation.

It would be great to add some summary data at the end of the manuscript, bringing (normalized?) key data of all mutations studied together so the readership can evaluate the defining differences in synaptic parameters in a single figure/panel.

We followed the reviewer's proposal and prepared Extended Data Figure 8. In this figure, we provide (i) an illustration of the main active zone proteins that have been linked to the group of SNAREopathies (Extended Data Figure 8a), (ii) a plot depicting the spectrum of pathogenic variants, patterns of inheritance and associated phenotypes reported for genes encoding neuronal SNAREs and associated AZ proteins, with frequency of diverse disease-causing variants and their patterns of inheritance (Extended Data Figure 8b), (iii) a short summary of the key genetic and clinical findings for each disease subtype (Extended Data Figure 8c), and (iv) a histogram plotting summarized key data for each tested variant (Extended Data Figure 8d). We trust that this figure provides an easily accessible overview about the UNC13A condition to the broad readership, and a guide for the experimental neuroscientists as well as geneticists, that can facilitate the accurate classification of newly-identified variants.

(4) The discussion is currently oriented towards molecular mechanisms and delivers relatively little scope (except the comparison between effects of UNC13A mutations early and later in life, which is great but also long). It would be great to touch upon a few more general topics, e.g. (i) genotype-phenotype relations (see above), (ii) is the classification into three groups definitive or are more classes predicted to emerge based on what we know about UNC13A function and the fact that currently known mutations are both in tolerant and intolerant regions? (iii) a comparison to developmental disorders caused by mutations in genes that operate in the same molecular machine as UNC13A and their fascinating genetic complexity.

We re-wrote large parts of the discussion to address the reviewer's criticism, acknowledging that the detailed description of the synapse physiology can be shortened. The discussion now includes (according to this reviewer's suggestions):

- references to genotype-phenotype relations, where we emphasize the hallmark presentation for each disease subtype (Lines 376-414), and
- a paragraph on additional disease subtypes we still anticipate to identify based on the known Munc13 biology (Lines 415-426): **We anticipate that additional UNC13A pathogenic variants and disease subtypes will emerge. In particular, loss of function without loss of protein is expected for variants that interfere with UNC13A function in SNARE complex assembly (e.g. the MUN or C2C domain), which may lead to a Type A-like condition. Based on AlphaMissense**

scores and the rareness of variants in the UNC13A hinge region (frequency in gnomADv4.1 database), almost all possible hinge amino acid exchanges (42/45=93%) show pathogenic predictions, and we anticipate additional pathogenic hinge variants to be identified as causing the Type B condition. Moreover, UNC13A hyperfunction has been observed in several structure-function studies¹¹⁻¹³, and variants in these regions may also result in a Type B condition. Finally, missense or in-frame variants in other critical domains involved in UNC13A regulation, including in the Ca²⁺-calmodulin binding sequence (aa 446-466)¹⁴ and in the Ca²⁺-phospholipid binding residues of the C2B domain¹⁵ may cause a Type C condition.

In addition, the new Extended Data Figure 8 includes

- (iii) a plot depicting the Spectrum of pathogenic variants, patterns of inheritance and associated phenotypes reported for genes encoding neuronal SNAREs and associated AZ proteins, with frequency of diverse disease-causing variants and their patterns of inheritance (Extended Data Figure 8b).

We thank this reviewer again for the constructive criticism and insightful comments, which we believe, we were able to fully address.

Reviewer #2:

Remarks to the Author:

The (many) authors of an international collaboration went through a concerted effort which resulted in describing a novel neurodevelopmental syndrome(s) linked to germline coding or splice site variants in the UNC13A gene, which can follow autosomal dominant (de novo or inherited heterozygous pathogenic variants) or autosomal recessive (biallelic pathogenic variants) inheritance patterns. The syndrome presents with a spectrum of symptoms, including developmental delay or intellectual disability, epilepsy, tremor, dyskinetic movements, and, in severe cases, early childhood mortality.

They use a broad array of assays to evaluate UNC13A protein stability and abundance at synapses, the strength and plasticity of neurotransmitter release using electrophysiological recordings in mouse hippocampal excitatory neurons, movement in *C. elegans* knock-in worms, synaptic responses to second messenger signaling, and structural changes through molecular dynamic modeling. Thus, they describe three distinct pathogenic mechanisms:

- Reduction in synaptic strength due to decreased UNC13A protein levels
- Increased neurotransmission resulting from a gain of function in UNC13A
- Impaired synaptic response to second messenger signaling

Based on this analysis (genotype, phenotype, physiological phenotype), they further classify UNC13A syndrome subtypes.

I am in the rare situation of having little to nothing to criticize here. The identified mutations significantly advance our understanding of UNC13A biology in humans and might potentially lay a foundation for developing therapeutic interventions. Clearly, this material apart from its clinical implications will also be of importance concerning our mechanistic understanding of Unc13 function in a generic manner. This also given that a bulk of the physiological information is retrieved from autaptic recordings, and much remains to be learned concerning the truly in vivo impact of these mutations (e.g. in the transmission across frequency space). Equally, it will (in the future) be important to address at which synapses these mutations ultimately execute their pathologically relevant dysfunctions. Still, given the major progress this paper marks, I do unreservedly suggest publication of the manuscript in its current form.

We are grateful to this reviewer for the positive evaluation of our work. This type of in-depth analysis and mechanistic investigation benefits from a great many studies in the field, which sharpened our

understanding of UNC13A function, and we are in debt to many synapse biology laboratories that paved the way for our study.

Reviewer #3:

Remarks to the Author:

This is very interesting study showing that pathogenic variants in UNC13A are (1) either biallelic or monoallelic with de novo or familial inheritance and (2) that pathogenic variants affect different UNC13A protein domains that participate in defining different properties of synaptic transmission, with evidence for at least three independent mechanisms: loss of function, gain-of-function (increased synaptic transmission) and dysregulation of transmission.

In OMIM two early onset syndromes are linked to UNC13A variants, albeit not uncontroversial: a congenital myasthenic syndrome and a dyskinetic movement disorder associated with delayed development and behavioral abnormalities. Could the authors discuss how their reported syndromes relate to these prior reported syndromes, and if not, how we should interpret these prior reports?

Indeed, few publications describing patients with UNC13A variations have been published, giving rise to the OMIM entries. A thorough and detailed examination of a patient with a premature homozygous stop codon by Engel et al.,¹⁰ led to the diagnosis of a congenital myasthenic syndrome, which implies that the disorder is at least in part of neuromuscular nature. This diagnosis is consistent with current knowledge on UNC13A expression at the neuromuscular junction¹⁶. In our Type A patients, we indeed observed symptoms that agree with the finding that complete or partial loss of UNC13A can result in expression of myasthenic symptoms. We have now added a sentence in the corresponding section of the Discussion to emphasize this connection: **Based on a previously reported case¹⁰, harbouring a homozygous stop codon variant and diagnosed with fatal myasthenia based on electromyography findings, and based on our patients 3 and 7 (Supplementary table 1, Extended data figure 2), we suggest considering myasthenic presentation in Type A patients. To what extent this is a typical presentation requires further investigations.** (lines 383-388).

The report of a single patient with a dyskinetic movement disorder, delayed development and behavioural abnormalities, refers to our previous work. In the present study, we now identified thirteen patients with variants in the same region (UNC13A hinge; aa 808-814, Type B) causing similar symptoms. This allows us to conclude that **A hallmark of the Type B condition is ataxia and tremor or dyskinetic movements (Fig. 1e and⁹)** (lines 399-400).

We accentuated this point more clearly in the Introduction and Discussion sections of the manuscript, and also illustrate this in the new overview figure (Extended Data Figure 8). Taken together, we see no contradiction between the available clinical descriptions so far and our new findings. In fact, our present study validates them in the context of a much larger cohort of patients, which enabled us to propose a sub-classification of the UNC13A syndrome.

The authors frequently refer to the recent ALS/FTD literature on UNC13A cryptic exon splicing induced by TDP-43 mislocalisation. Indeed, the question is, if this study helps to understand the mechanisms at play in ALS/FTD, since this has not yet been resolved. The cryptic exon is located between exon 20 and 21, so sort of right in the "UNC13A Hinge" as the authors call it. Still, we "think" the effect in ALS/FTD is "loss of function" as UNC13A is misspliced and the aberrant mRNA species are degraded by non-sense mediated decay. Nevertheless, we know that in ALS/FTD glutamate excitotoxicity is one of many pathogenic mechanisms, so I wondered if the authors could reflect on which of their presented mechanisms could be at play in ALS/FTD? It is very intriguing to see that coding mutations in UNC13A could lead to a toxic gain of increased synaptic transmission, but how can one this reconcile with the presumed loss of function following missplicing of UNC13A? It would be great if the authors could reflect on this, and dedicate a paragraph to this.

We believe the major findings that are relevant for a better understanding of the role of UNC13A in ALS/FTD concern the Type A condition, in which reduced UNC13A levels are observed. We report that (i) UNC13A haploinsufficiency in humans is well-tolerated. This is consistent with previous experimental findings indicating that (ii) UNC13A heterozygosity is fully-tolerated in mice, with no changes in synaptic transmission even though UNC13A expression is reduced to about 50% of WT levels³. Here, we experimentally determined (iii) a reduction of UNC13A expression to 20-30% of WT levels and aberrant patterns of synaptic transmission and short-term plasticity in cultured hippocampal neurons expressing UNC13A^{R202H} (Figs 2, 3, 5, 6), and, importantly, (iv) that UNC13A^{R202H} severely interferes with human motor function (Supplementary table 1). Taken together, these data allow us to suggest that declining UNC13A expression in neurons affected by ALS/FTD may cause aberrant patterns of synaptic transmission. Based on the above, we hypothesize that therapeutic approaches that stabilize UNC13A expression at about 50% of the respective WT levels (but not necessarily 100%) are desirable. Moreover, we also report that (v) UNC13A^{R202H}-expressing neurons maintain about 60% of the synaptic strength as compared to healthy neurons, and that (vi) even low UNC13A expression levels (20-30%) are clearly beneficial in comparison to a complete loss of expression, as patients expressing the R202H variant (and patients with (possibly leaky) splice site variants) have higher chances of survival as compared to patients carrying a homozygous premature stop codon in the *UNC13A* gene - who die in early childhood^{10,17}. We discuss these data to indicate that *any* increase in the levels of UNC13A could be advantageous to patients, with the hope that this will be constructive to any initiative aimed at developing UNC13-targeting pharmacology for ALS/FTD patients.

This reviewer correctly points out the physical proximity between the UNC13A hinge sequence (exon 20), variations in which cause a Type B UNC13A condition, and the single nucleotide polymorphisms identified to confer ALS/FTD risk (intron sequence between exons 20 and 21). We do not have a mechanistic explanation for this link, nor a hypothesis that is well-supported by current literature. It is possible that this region of the *UNC13A* gene is particularly susceptible to genetic alterations, and it would be fascinating to explore this possibility. Finally, we fully agree with this reviewer that the UNC13A gain of function may also result in excitotoxicity, a mechanism that has been extensively discussed as promoting nerve cell degeneration in ALS/FTD. However, patients carrying UNC13A hinge variations do not show signs of increased neurodegeneration up to an age of thirty years. We will continue to follow these patients as they grow older to monitor whether such a phenotype may arise. In the revised manuscript however, we refrain from making the statement that UNC13A gain of function does not result in excitotoxicity.

Lines 464-480: The characterization of UNC13A^{R202H}-expressing neurons revealed that expression levels of 20-30% of WT levels strongly impair synaptic function, particularly the pattern of plasticity and the responses to DAG/PDBu (Figs. 2, 3, 5, 6). Although we cannot exclude that R202H has additional effects on UNC13A function beyond reducing its expression levels, these data suggest that UNC13A levels below 50% profoundly change synaptic transmission and plasticity properties in hippocampal neurons in culture¹⁸, and severely interfere with human motor function, as seen in all Type A patients (Fig. 1e). This raises the possibility that such an altered neurotransmission pattern could accompany the cellular pathology in ALS/FTD as UNC13A levels decline, and potentially even exacerbate disease symptoms. However, we also note that 20-30% of UNC13A expression levels are already sufficient to support ~60% of neurotransmitter release in cultured neurons during low AP frequencies, and that patients with low expression of UNC13A have improved chances of survival as compared to patients with no functional UNC13A, who die in early childhood^{10,17}. Taken together, we propose that therapeutic strategies that stabilize even a minimal level of UNC13A expression may already be beneficial, and, because there are no indications for pathological consequences of UNC13A haploinsufficiency in humans (this study and few published cases^{17, 10}) or in heterozygous mice³, that restoration of UNC13A levels approaching 50% is a sufficient therapeutic target.

I have one methodological question: how exactly did the authors quantify synapse numbers and in which (non)human systems? This is a bit unclear from the methods section. Is it done using antibodies against SHANK2 and VGLUT1? In *C. elegans*?

Synapse numbers were determined in murine autaptic hippocampal cultures. We stained neurons with antibodies against Shank2, VGLUT1, Map2 and Munc13-1, and imaged entire neurons. Following established procedures, synapses were defined as puncta exhibiting co-localization of Shank2 and VGLUT1 immunofluorescence at a restricted distance from the MAP2 signal (Fig. 2D). We then determined the fraction of synapses showing expression of UNC13A (Fig. 2E). Next, we measured the immunofluorescence intensities of UNC13A within all puncta identified at synapses for a given neuron. For each condition, the average immunofluorescence intensity distribution is shown (Fig. 2F). We revised the main text, figure legends, and Materials and Methods section accordingly to improve clarity.

Textual:

The introduction has a strange start with "In a ground-breaking screen for genes controlling *C. elegans* behaviour¹, uncoordinated (UNC) strain number 13 (unc-13) was scored as severely affected." I would start with a combination of the second sentence, e.g. In a groundbreaking... worms with a"

These sentences have now been revised and shortened to comply with manuscript length requirements.

Bibliography

1. Augustin, I., Betz, A., Herrmann, C., Jo, T., and Brose, N. (1999). Differential expression of two novel Munc13 proteins in rat brain. *Biochem J* 337 (Pt 3), 363-371.
2. Koch, H., Hofmann, K., and Brose, N. (2000). Definition of Munc13-homology-domains and characterization of a novel ubiquitously expressed Munc13 isoform. *Biochem J* 349, 247-253. 10.1042/0264-6021:3490247.
3. Augustin, I., Rosenmund, C., Südhof, T.C., and Brose, N. (1999). Munc13-1 is essential for fusion competence of glutamatergic synaptic vesicles. *Nature* 400, 457-461. 10.1038/22768.
4. Varoqueaux, F., Sigler, A., Rhee, J.S., Brose, N., Enk, C., Reim, K., and Rosenmund, C. (2002). Total arrest of spontaneous and evoked synaptic transmission but normal synaptogenesis in the absence of Munc13-mediated vesicle priming. *Proc Natl Acad Sci U S A* 99, 9037-9042. 10.1073/pnas.122623799.
5. Sigler, A., Oh, W.C., Imig, C., Altas, B., Kawabe, H., Cooper, B.H., Kwon, H.B., Rhee, J.S., and Brose, N. (2017). Formation and Maintenance of Functional Spines in the Absence of Presynaptic Glutamate Release. *Neuron* 94, 304-311 e304. 10.1016/j.neuron.2017.03.029.
6. Lipstein, N., Schaks, S., Dimova, K., Kalkhof, S., Ihling, C., Kolbel, K., Ashery, U., Rhee, J., Brose, N., Sinz, A., and Jahn, O. (2012). Nonconserved Ca(2+)/calmodulin binding sites in Munc13s differentially control synaptic short-term plasticity. *Mol Cell Biol* 32, 4628-4641. 10.1128/MCB.00933-12.
7. Kawabe, H., Mitkovski, M., Kaeser, P.S., Hirrlinger, J., Opazo, F., Nestvogel, D., Kalla, S., Fejtova, A., Verrier, S.E., Bungers, S.R., et al. (2017). ELKS1 localizes the synaptic vesicle priming protein bMunc13-2 to a specific subset of active zones. *J Cell Biol* 216, 1143-1161. 10.1083/jcb.201606086.
8. Bohme, M.A., Beis, C., Reddy-Alla, S., Reynolds, E., Mampell, M.M., Grasskamp, A.T., Lutzkendorf, J., Bergeron, D.D., Driller, J.H., Babikir, H., et al. (2016). Active zone scaffolds differentially accumulate Unc13 isoforms to tune Ca(2+) channel-vesicle coupling. *Nat Neurosci* 19, 1311-1320. 10.1038/nn.4364.
9. Lipstein, N., Verhoeven-Duif, N.M., Michelassi, F.E., Calloway, N., van Hasselt, P.M., Pienkowska, K., van Haften, G., van Haelst, M.M., van Empelen, R., Cuppen, I., et al. (2017). Synaptic UNC13A protein variant causes increased neurotransmission and dyskinetic movement disorder. *J Clin Invest* 127, 1005-1018. 10.1172/JCI90259.
10. Engel, A.G., Selcen, D., Shen, X.M., Milone, M., and Harper, C.M. (2016). Loss of MUNC13-1 function causes microcephaly, cortical hyperexcitability, and fatal myasthenia. *Neurol Genet* 2, e105. 10.1212/NXG.000000000000105.
11. Li, L., Liu, H., Hall, Q., Wang, W., Yu, Y., Kaplan, J.M., and Hu, Z. (2019). A Hyperactive Form of unc-13 Enhances Ca(2+) Sensitivity and Synaptic Vesicle Release Probability in *C. elegans*. *Cell Rep* 28, 2979-2995 e2974. 10.1016/j.celrep.2019.08.018.
12. Michelassi, F., Liu, H., Hu, Z., and Dittman, J.S. (2017). A C1-C2 Module in Munc13 Inhibits Calcium-Dependent Neurotransmitter Release. *Neuron* 95, 577-590 e575. 10.1016/j.neuron.2017.07.015.
13. Camacho, M., Basu, J., Trimbuch, T., Chang, S., Pulido-Lozano, C., Chang, S.S., Duluvova, I., Abo-Rady, M., Rizo, J., and Rosenmund, C. (2017). Heterodimerization of Munc13 C2A domain with RIM regulates synaptic vesicle docking and priming. *Nat Commun* 8, 15293. 10.1038/ncomms15293.
14. Junge, H.J., Rhee, J.S., Jahn, O., Varoqueaux, F., Spiess, J., Waxham, M.N., Rosenmund, C., and Brose, N. (2004). Calmodulin and Munc13 form a Ca2+ sensor/effector complex that controls short-term synaptic plasticity. *Cell* 118, 389-401. 10.1016/j.cell.2004.06.029.
15. Shin, O.H., Lu, J., Rhee, J.S., Tomchick, D.R., Pang, Z.P., Wojcik, S.M., Camacho-Perez, M., Brose, N., Machius, M., Rizo, J., et al. (2010). Munc13 C2B domain is an activity-dependent Ca2+ regulator of synaptic exocytosis. *Nat Struct Mol Biol* 17, 280-288. 10.1038/nsmb.1758.
16. Varoqueaux, F., Sons, M.S., Plomp, J.J., and Brose, N. (2005). Aberrant morphology and residual transmitter release at the Munc13-deficient mouse neuromuscular synapse. *Mol Cell Biol* 25, 5973-5984. 10.1128/MCB.25.14.5973-5984.2005.
17. Mullins, J.R., McFadden, K., Snow, N., and Oviedo, A. (2022). Homozygous UNC13A Variant in an Infant With Congenital Encephalopathy and Severe Neuromuscular Phenotype: A Case Report With Detailed Central Nervous System Neuropathologic Findings. *Cureus* 14, e30774. 10.7759/cureus.30774.
18. Zarebidaki, F., Camacho, M., Brockmann, M.M., Trimbuch, T., Herman, M.A., and Rosenmund, C. (2020). Disentangling the Roles of RIM and Munc13 in Synaptic Vesicle Localization and Neurotransmission. *J Neurosci* 40, 9372-9385. 10.1523/JNEUROSCI.1922-20.2020.

NG-A65794R - POINT-BY-POINT REPLY TO THE REVIEWERS' COMMENTS

A Novel Neurodevelopmental Syndrome Caused by Pathogenic *UNC13A* Variants that Impair Synaptic Function (Asadollahi et al.)

We would like to thank all reviewers again for their positive assessment of our work.

Reviewer #1 (Remarks to the Author):

the authors provide adequate responses to all my prior concerns and made adequate adjustment to the manuscript. The authors deserve a big complement on this excellent study

We thank this reviewer for the support and for the complements

Reviewer #2 (Remarks to the Author):

I am impressed by the revision and suggest publication unreservedly.

We appreciate the support and time invested in assessing our manuscript.

Reviewer #3 (Remarks to the Author):

The authors have addressed all reviewers comments more than adequately, I appreciate the additions in the Discussion further placing the findings into context.

I only noticed on typo: Extended data Figure 8: the table says "Inheritance", needs to be "Inheritance". Very well done all in all.

We appreciate the attentive review and the great suggestion that drove the generation of the overview figure. The spelling mistakes in the figure have been corrected.